# Biophysical mechanisms in the mammalian respiratory oscillator re-examined with a new data-driven computational model

Ryan S Phillips[1,2], Tibin T John[1], Hidehiko Koizumi[1], Yaroslav I Molkov[3,4], Jeffrey C Smith[1]*

[1]Cellular and Systems Neurobiology Section, National Institute of Neurological Disorders and Stroke, National Institutes of Health, Bethesda, United States; [2]Department of Physics, University of New Hampshire, Durham, United States; [3]Department of Mathematics and Statistics, Georgia State University, Atlanta, United States; [4]Neuroscience Institute, Georgia State University, Atlanta, United States

**Abstract** An autorhythmic population of excitatory neurons in the brainstem pre-Bötzinger complex is a critical component of the mammalian respiratory oscillator. Two intrinsic neuronal biophysical mechanisms—a persistent sodium current ($I_{NaP}$) and a calcium-activated non-selective cationic current ($I_{CAN}$)—were proposed to individually or in combination generate cellular- and circuit-level oscillations, but their roles are debated without resolution. We re-examined these roles in a model of a synaptically connected population of excitatory neurons with $I_{CAN}$ and $I_{NaP}$. This model robustly reproduces experimental data showing that rhythm generation can be independent of $I_{CAN}$ activation, which determines population activity amplitude. This occurs when $I_{CAN}$ is primarily activated by neuronal calcium fluxes driven by synaptic mechanisms. Rhythm depends critically on $I_{NaP}$ in a subpopulation forming the rhythmogenic kernel. The model explains how the rhythm and amplitude of respiratory oscillations involve distinct biophysical mechanisms.
DOI: https://doi.org/10.7554/eLife.41555.001

*For correspondence:
smithj2@ninds.nih.gov

**Competing interests:** The authors declare that no competing interests exist.

## Introduction

Defining cellular and circuit mechanisms generating the vital rhythm of breathing in mammals remains a fundamental unsolved problem of wide-spread interest in neurophysiology (*Richter and Smith, 2014*; *Del Negro et al., 2018*; *Ramirez and Baertsch, 2018a*), with potentially far-reaching implications for understanding mechanisms of oscillatory circuit activity and rhythmic motor pattern generation in neural systems (*Marder and Calabrese, 1996*; *Buzsaki, 2006*; *Grillner, 2006*; *Kiehn, 2006*). The brainstem pre-Bötzinger complex (pre-BötC) region (*Smith et al., 1991*) located in the ventrolateral medulla oblongata is established to contain circuits essential for respiratory rhythm generation (*Smith et al., 2013*; *Del Negro et al., 2018*), but the operational cellular biophysical and circuit synaptic mechanisms are continuously debated. Pre-BötC excitatory neurons and circuits have autorhythmic properties and drive motor circuits that can be isolated and remain rhythmically active in living rodent brainstem slices in vitro. Numerous experimental and theoretical analyses have focused on the rhythmogenic mechanisms operating in these in vitro conditions to provide insight into biophysical and circuit processes involved, with potential relevance for rhythm generation during breathing in vivo (*Feldman and Del Negro, 2006*; *Lindsey et al., 2012*; *Richter and Smith, 2014*; *Ramirez and Baertsch, 2018b*). The ongoing rhythmic activity in vitro has been suggested to arise from a variety of cellular and circuit biophysical mechanisms including from a subset(s) of intrinsically bursting neurons which, through excitatory synaptic interactions, recruit

and synchronize neurons within the network (pacemaker-network models) (*Butera et al., 1999b*; *Ramirez et al., 2004*; *Toporikova and Butera, 2011*; *Chevalier et al., 2016*), or as an emergent network property through recurrent excitation (e.g. *Rekling and Feldman, 1998*; *Jasinski et al., 2013*) and/or synaptic depression (group pacemaker model) (*Rubin et al., 2009a*; *Del Negro et al., 2010*).

From these previous analyses, involvement of two possible cellular-level biophysical mechanisms have been proposed. One based on a slowly inactivating persistent sodium current ($I_{NaP}$) (*Butera et al., 1999a*), and the other on a calcium-activated non-selective cation current ($I_{CAN}$) coupled to intracellular calcium ($[Ca]_i$) dynamics (for reviews see *Rybak et al., 2014*; *Del Negro et al., 2010*), or a combination of both mechanisms (*Thoby-Brisson and Ramirez, 2001*; *Jasinski et al., 2013*; *Peña et al., 2004*). Despite the extensive experimental and theoretical investigations of these sodium- and calcium-based mechanisms, the actual roles of $I_{NaP}$, $I_{CAN}$ and the critical source(s) of $[Ca]_i$ transients in the pre-BötC are still unresolved. Furthermore, in pre-BötC circuits, the process of rhythm generation must be associated with an amplitude of circuit activity sufficient to drive downstream circuits to produce adequate inspiratory motor output. Biophysical mechanisms involved in generating the amplitude of pre-BötC circuit activity have also not been established.

$I_{NaP}$ is proposed to mediate an essential oscillatory burst-generating mechanism since pharmacologically inhibiting $I_{NaP}$ abolishes intrinsic neuronal rhythmic bursting as well as pre-BötC circuit inspiratory activity and rhythmic inspiratory motor output in vitro (*Koizumi et al., 2008*; *Toporikova et al., 2015*), although some studies suggest that block of both $I_{NaP}$ and $I_{CAN}$ are necessary to disrupt rhythmogenesis in vitro (*Peña et al., 2004*). Theoretical models of cellular and circuit activity based on $I_{NaP}$-dependent bursting mechanisms closely reproduce experimental observations such as voltage-dependent frequency control, spike-frequency adaptation during bursts, and pattern formation of inspiratory motor output (*Butera et al., 1999b*; *Pierrefiche et al., 2004*; *Smith et al., 2007*). This indicates the plausibility of $I_{NaP}$-dependent rhythm generation.

In the pre-BötC, $I_{CAN}$ was originally postulated to underlie intrinsic pacemaker-like oscillatory bursting at the cellular level and contribute to circuit-level rhythm generation, since intrinsic bursting in a subset of neurons in vitro was found to be terminated by the $I_{CAN}$ inhibitor flufenamic acid (FFA) (*Peña et al., 2004*). Furthermore, inhibition of $I_{CAN}$ in the pre-BötC reduces the amplitude of the rhythmic depolarization (inspiratory drive potential) driving neuronal bursting and can eliminate inspiratory motor activity in vitro (*Pace et al., 2007*). $I_{CAN}$ became the centerpiece of the 'group pacemaker' model for rhythm generation, in which this conductance was proposed to be activated by inositol trisphosphate (IP3) receptor/ER-mediated intracellular calcium fluxes initiated via glutamatergic metabotropic receptor-mediated signaling in the pre-BötC excitatory circuits (*Del Negro et al., 2010*). The molecular correlate of $I_{CAN}$ was postulated to be the transient receptor potential channel M4 (TRPM4) (*Mironov, 2008*; *Pace et al., 2007*), one of the two known $Ca^{2+}$-activated TRP channels (*Guinamard et al., 2010*; *Ullrich et al., 2005*), or alternatively, by the transient receptor potential channels C3/7 (TRPC3/7) (*Ben-Mabrouk and Tryba, 2010*); however, these latter channels are not known to be $Ca^{2+}$-activated (*Clapham, 2003*). TRPM4 and TRPC3 have now been identified by immunolabeling and RNA expression profiling in pre-BötC inspiratory neurons in vitro (*Koizumi et al., 2018*).

Investigations into the sources of intracellular $Ca^{2+}$ activating $I_{CAN}$/TRPM4 suggested that (1) somatic calcium transients from voltage-gated sources do not contribute to the inspiratory drive potential (*Morgado-Valle et al., 2008*), (2) IP3/ER-mediated intracellular $Ca^{2+}$ release does not contribute to inspiratory rhythm generation in vitro, and (3) in the dendrites calcium transients may be triggered by excitatory synaptic inputs and travel in a wave propagated to the soma (*Mironov, 2008*). Theoretical studies have demonstrated the plausibility of $[Ca]_i$-$I_{CAN}$-dependent bursting (*Rubin et al., 2009b*; *Toporikova and Butera, 2011*); however, these models omit $I_{NaP}$ and/or depend on additional unproven mechanisms to generate intracellular calcium oscillations to provide burst termination, such as IP3-dependent calcium-induced calcium release (*Toporikova and Butera, 2011*), partial depolarization block of action potentials (*Rubin et al., 2009a*), and the $Na^+$/$K^+$ pump (*Jasinski et al., 2013*). Interestingly, pharmacological inhibition of $I_{CAN}$/TRPM4 has been shown to produce large reductions in the amplitude of pre-BötC inspiratory neuron population activity with essentially no, or minor perturbations of inspiratory rhythm (*Peña et al., 2004*). These observations constrain the role of $I_{CAN}$, and require theoretical re-examination of pre-BötC neuronal conductance

mechanisms and network dynamics, particularly how rhythm generation mechanisms can be independent of $I_{CAN}$-dependent mechanisms that regulate the amplitude of network activity.

In this theoretical study, we examine the role of $I_{CAN}$ in pre-BötC excitatory circuits by considering two plausible mechanisms of intracellular calcium fluxes: (1) from voltage-gated and (2) from synaptically activated sources. We deduce that $I_{CAN}$ is primarily activated by calcium transients that are coupled to rhythmic excitatory synaptic inputs originating from $I_{NaP}$-dependent bursting inspiratory neurons. Additionally, we show that $I_{CAN}$ contributes to the inspiratory drive potential by mirroring the excitatory synaptic current. This concept is consistent with a mechanism underlying generation of the inspiratory drive potential involving a synaptic-based $I_{CAN}$ activation described in previous work (*Pace et al., 2007*; *Rubin et al., 2009a*). Our model explains the experimental observations obtained from in vitro neonatal rodent slices isolating the pre-BötC, showing large reductions in network activity amplitude by inhibiting $I_{CAN}$/TRPM4 without perturbations of inspiratory rhythm generation in pre-BötC excitatory circuits in vitro. The model supports the concept that $I_{CAN}$ activation in a subpopulation of pre-BötC excitatory neurons is critically involved in amplifying synaptic drive from a subset of neurons whose rhythmic bursting is critically dependent on $I_{NaP}$ and forms the kernel for rhythm generation in vitro. The model suggests how the functions of generating the rhythm and amplitude of inspiratory oscillations in pre-BötC excitatory circuits are determined by distinct biophysical mechanisms.

## Results

### $\bar{g}_{CAN}$ variation has opposite effects on amplitude and frequency of network bursting in the $Ca_V$ and $Ca_{Syn}$ models

Experimental work (*Peña et al., 2004*) has demonstrated that pharmacological inhibition of $I_{CAN}$/TRPM4 in the pre-BötC from in vitro neonatal mouse/rat slice preparations, strongly reduces the amplitude of (or completely eliminates) the inspiratory hypoglossal (XII) motor output, as well as the amplitude of pre-BötC excitatory circuit activity that is highly correlated with the decline of XII activity, while having relatively little effect on inspiratory burst frequency. Here, we systematically examine in our model the relationship between $I_{CAN}$ conductance ($\bar{g}_{CAN}$) and the amplitude and frequency of network activity for voltage-gated ($Ca_V$) and synaptically activated sources ($Ca_{Syn}$) of intracellular calcium in a heterogeneous network of 100 synaptically coupled single-compartment pre-BötC model excitatory neurons. In addition to voltage-gated and synaptically activated calcium currents, each model neuron incorporates voltage-gated action potential generating currents, as well as $I_{CAN}$, $I_{NaP}$, leak, and excitatory synaptic currents adapted from the conductance-based biophysical model of *Jasinski et al. (2013)* (see Materials and methods for a full model description). We note that in our full model, $I_{CAN}$ is activated by both voltage-gated and synaptic mechanisms, consistent with experimental results (*Thoby-Brisson and Ramirez, 2001*; *Pace et al., 2007*; *Morgado-Valle et al., 2008*). Initial separate consideration of the $Ca_V$ and $Ca_{Syn}$ provides a means to deduce the relative contribution of these two general sources of intracellular calcium to $I_{CAN}$ activation. We found that reduction of $\bar{g}_{CAN}$ drives opposing effects on network activity amplitude (spike/s/neuron) and frequency that are dependent on the source of intracellular calcium transients (*Figure 1*). The network activity amplitude is a measure of the average neuronal population firing rate and is defined by the number of spikes generated by the network per 50 ms bin divided by the number of neurons in the network. In the $Ca_V$ network, where calcium influx is generated exclusively from voltage-gated calcium channels, increasing $\bar{g}_{CAN}$ has no effect on amplitude but increases the frequency of network oscillations (*Figure 1A, C, D*). Conversely, in the $Ca_{Syn}$ network where calcium influx is generated exclusively by excitatory synaptic input, increasing $\bar{g}_{CAN}$ strongly increases the amplitude and slightly decreases the oscillation frequency (*Figure 1B, C, D*).

### Effects of subthreshold activation of $I_{CAN}$ on network frequency

In $I_{NaP}$-dependent bursting neurons in the pre-BötC, bursting frequency depends on their excitability (i.e. baseline membrane potential) which can be controlled in different ways, for example by directly injecting a depolarizing current (*Smith et al., 1991*; *Del Negro et al., 2005*; *Yamanishi et al., 2018*) or varying the conductance and/or reversal potentials of some ionic channels (*Butera et al., 1999a*).

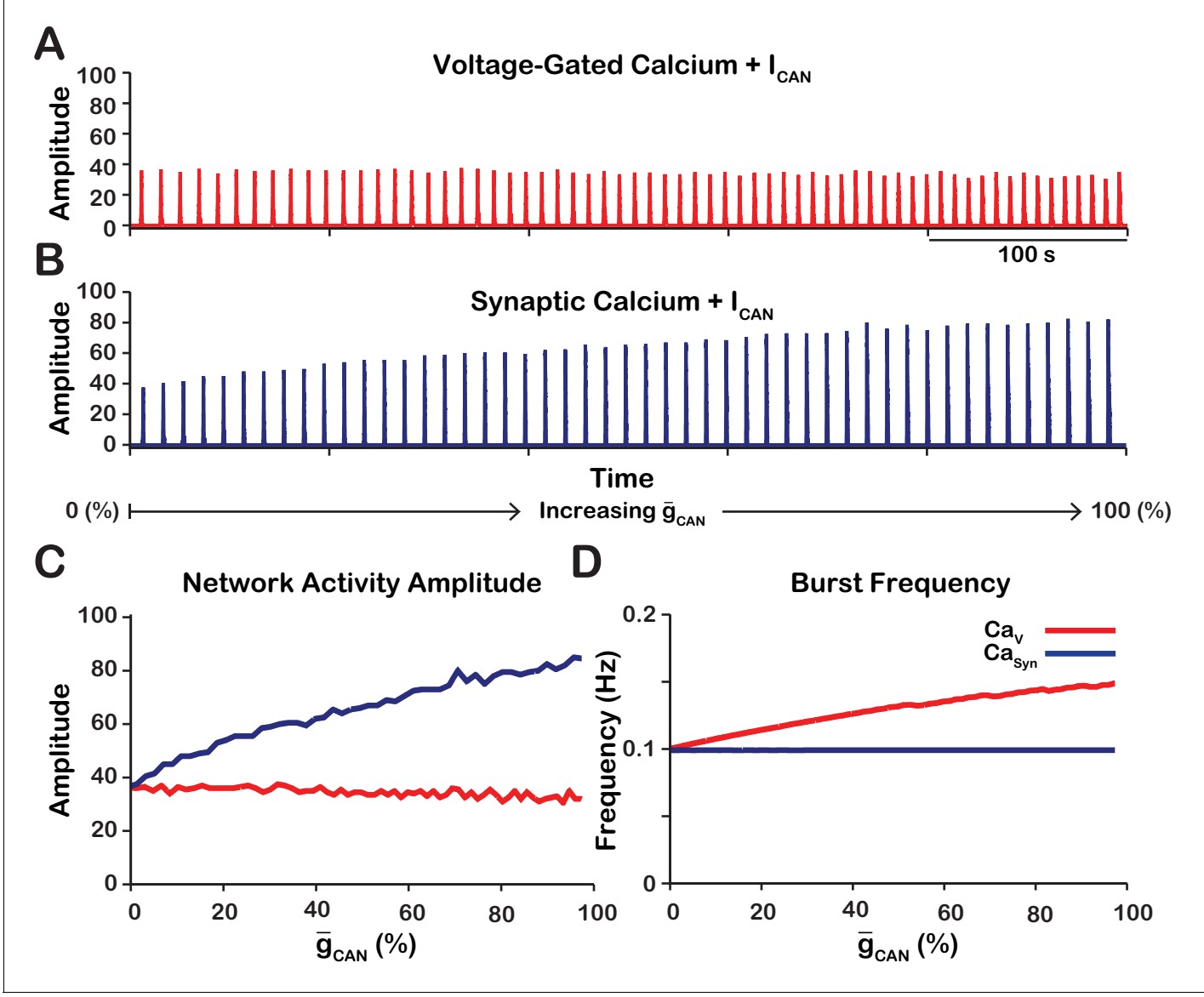

**Figure 1.** Manipulations of $\bar{g}_{CAN}$ in the $\boldsymbol{Ca_V}$ and $\boldsymbol{Ca_{Syn}}$ networks produce opposite effects on network activity amplitude (spikes/s) and frequency. (A and B) Histograms of neuronal population activity amplitude in the $\boldsymbol{Ca_V}$, and $\boldsymbol{Ca_{Syn}}$ networks with linearly increasing $\bar{g}_{CAN}$. (C) Plot of $\bar{g}_{CAN}$ (% of the baseline mean value for the simulated population) vs. network activity amplitude for the $\boldsymbol{Ca_V}$ and $\boldsymbol{Ca_{Syn}}$ networks in A and B. (D) Plot of $\bar{g}_{CAN}$ (%) vs. network frequency for the $\boldsymbol{Ca_V}$ and $\boldsymbol{Ca_{Syn}}$ networks in A and B. $\boldsymbol{Ca_V}$ network parameters: $\bar{g}_{Ca} = 1.0$ ($\boldsymbol{nS}$), $\boldsymbol{P_{Ca}} = 0.0$, $\boldsymbol{P_{Syn}} = 0.05$ and $\boldsymbol{W_{max}} = 0.2$ ($\boldsymbol{nS}$). $\boldsymbol{Ca_{Syn}}$ network parameters: $\bar{g}_{Ca} = 0$ ($\boldsymbol{nS}$), $\boldsymbol{P_{Ca}} = 0.01$, $\boldsymbol{P_{Syn}} = 0.05$ and $\boldsymbol{W_{max}} = 0.2$ ($\boldsymbol{nS}$).

DOI: https://doi.org/10.7554/eLife.41555.002

Due to their relatively short duty cycle, the bursting frequency in these neurons is largely determined by the interburst interval, defined as the time between the end of one burst and the start of the next. During the burst, $I_{NaP}$ slowly inactivates (*Butera et al., 1999a*) resulting in burst termination and abrupt neuronal hyperpolarization. The interburst interval is then determined by the amount of time required for $I_{NaP}$ to recover from inactivation and return the membrane potential back to the threshold for burst initiation. This process is governed by the kinetics of $I_{NaP}$ inactivation gating variable $h_{NaP}$. Higher neuronal excitability reduces the value of $h_{NaP}$ required to initiate bursting. Consequently, the time required to reach this value is decreased, which results in a shorter interburst interval and increased frequency.

To understand how changing $\bar{g}_{CAN}$ affects network bursting frequency, we quantified the values of $h_{NaP}$ averaged over all rhythm-generating pacemaker neurons immediately preceding each network burst and, also, the average $I_{CAN}$ values between the bursts in the $Ca_V$ and $Ca_{syn}$ networks (**Figure 2**). In the $Ca_V$ network, $I_{Ca}$ as modeled remains residually activated between the bursts thus creating the background calcium concentration which partially activates $I_{CAN}$. Therefore, between the bursts $I_{CAN}$ functions as a depolarizing leak current. Consistently, we found that in the $Ca_V$ network increasing $\bar{g}_{CAN}$ increases $I_{CAN}$ (**Figure 2A**), progressively depolarizing the network, which reduces the $h_{NaP}$ threshold for burst initiation (**Figure 2B**) and, thus, increases network oscillation frequency (**Figure 1D**).

In the $Ca_{syn}$ model, the intracellular calcium depletes entirely during the interburst interval. Consequently, increasing $\bar{g}_{CAN}$ has no effect on $I_{CAN}$ (**Figure 2A**) and frequency is essentially unaffected (**Figure 1D**).

## Changes in network activity amplitude are driven by recruitment of neurons

As previously stated, the network activity amplitude is defined as the total number of spikes produced by the network per a time bin divided by the number of neurons in the network. Consequently, changes in amplitude can only occur by increasing the number of neurons participating in bursts (recruitment) and/or increasing the firing rate of the recruited neurons. To analyze changes in amplitude, we quantified the number of recruited neurons (**Figure 3A**) and the average spike frequency in recruited neurons (**Figure 3B**) as a function of $\bar{g}_{CAN}$ for both network models. In the $Ca_V$ network, increasing $\bar{g}_{CAN}$ increases the number of recruited neurons (**Figure 3A**), but decreases the average spiking frequency in recruited neurons (**Figure 3B**) which, together result in no change in amplitude (**Figure 1C**). In the $Ca_{Syn}$ network, increasing $\bar{g}_{CAN}$ strongly increases the number of recruited neurons (**Figure 3A**) and increases the spike frequency of recruited neurons (**Figure 3B**) resulting in a large increase in network activity amplitude (**Figure 3C**).

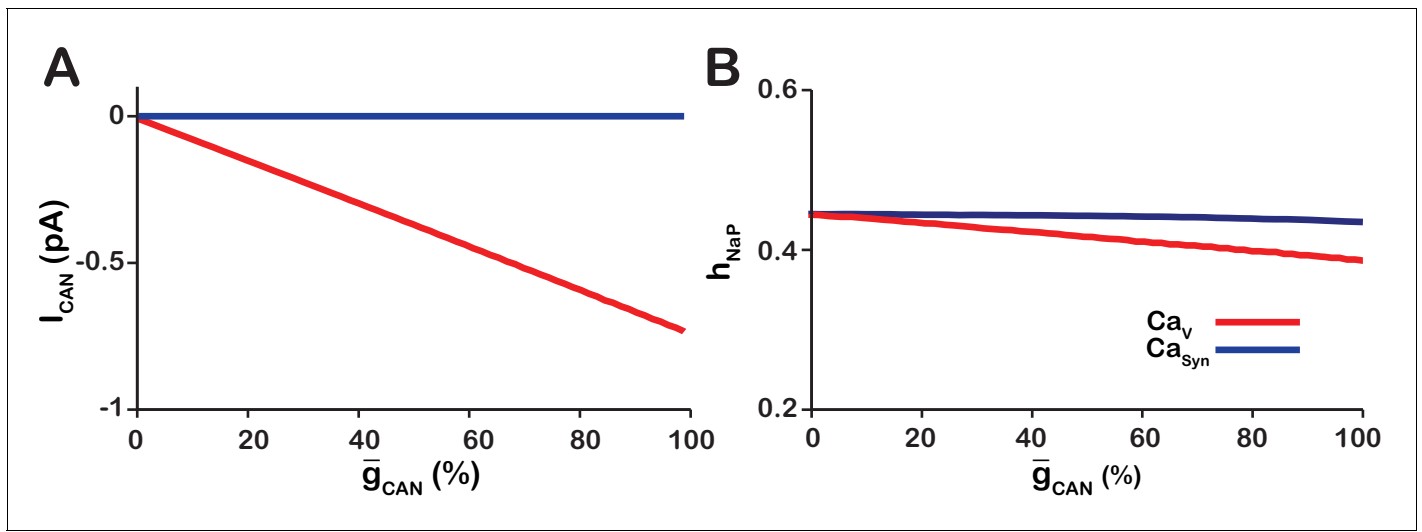

**Figure 2.** Calcium source and $\bar{g}_{CAN}$-dependent effects on cellular properties regulating network frequency for the simulations presented in **Figure 1**. (A) Average magnitude of $I_{CAN}$ in pacemaker neurons during the interburst interval for the $Ca_V$ (red) and $Ca_{Syn}$ (blue) networks. (B) Average inactivation ($h_{NaP}$) of the burst generating current $I_{NaP}$ in pacemaker neurons immediately preceding each network burst as a function of $\bar{g}_{CAN}$ (%) for the voltage-gated and synaptic calcium networks. $Ca_V$ network parameters: $\bar{g}_{Ca} = 1.0$ ($nS$), $P_{Ca} = 0.0$, $P_{Syn} = 0.05$ and $W_{max} = 0.2$ ($nS$). $Ca_{Syn}$ network parameters: $\bar{g}_{Ca} = 0$ ($nS$), $P_{Ca} = 0.01$, $P_{Syn} = 0.05$ and $W_{max} = 0.2$ ($nS$).
DOI: https://doi.org/10.7554/eLife.41555.003

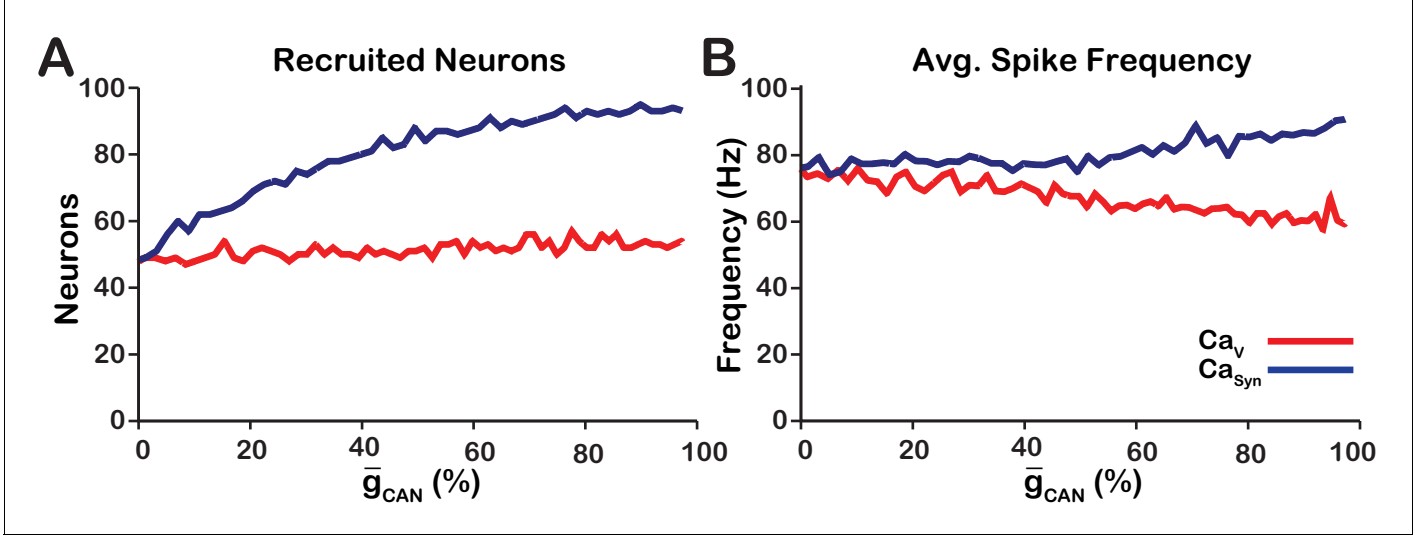

**Figure 3.** Calcium source and $\bar{g}_{CAN}$-dependent effects on cellular properties regulating network activity amplitude for the simulations presented in *Figure 1*. (A) Number of recruited neurons in the modeled population of 100 neurons as a function of $\bar{g}_{CAN}$ (%) for voltage-gated and synaptic calcium sources. The number of recruited neurons is defined as the peak number of spiking neurons per bin during a network burst. (B) Average spiking frequency of recruited neurons as a function of $\bar{g}_{CAN}$ for the voltage-gated and synaptic calcium mechanism. Average spiking frequency is defined the number of spikes per bin divided by the number of recruited neurons. The parameters used in these simulations are: $Ca_V$: $\bar{g}_{Ca} = 1.0$ ($nS$), $P_{Ca} = 0.0$, $P_{Syn} = 0.05$ and $W_{max} = 0.2$ ($nS$). $Ca_{Syn}$: $\bar{g}_{Ca} = 0$ ($nS$), $P_{Ca} = 0.01$, $P_{Syn} = 0.05$ and $W_{max} = 0.2$ ($nS$).
DOI: https://doi.org/10.7554/eLife.41555.004

## Manipulating $\bar{g}_{CAN}$ in the $Ca_{Syn}$ model is qualitatively equivalent to changing the strength of synaptic interactions

Since changes in $\bar{g}_{CAN}$ in the $Ca_{Syn}$ model primarily affect network activity amplitude through recruitment of neurons, and the network activity amplitude strongly depends on the strength of synaptic interactions, we next examined the relationship between $\bar{g}_{CAN}$, synaptic strength and network activity amplitude and frequency (*Figure 4*). Synaptic strength is defined as the number of neurons multiplied by the synaptic connection probability multiplied by the average weight of synaptic connections ($N \cdot P_{Syn} \cdot \frac{1}{2} W_{max}$), where the weight of synaptic connections $W_{max}$ ranges from 0.0 to 1.0 $nS$. We found that the effects of varying $\bar{g}_{CAN}$ or the synaptic strength on network activity amplitude and frequency are qualitatively equivalent in the $Ca_{Syn}$ network which is indicated by symmetry of the heat plots (across the X=Y line) in *Figure 4A,B*. This symmetry results from the fact that the effective strength of synaptic interactions in the network is roughly proportional to a product of the synaptic strength and $\bar{g}_{CAN}$. A transition from bursting to tonic spiking occurs when this effective excitation exceeds a certain critical value. This is why the bifurcation curve corresponding to a transition from rhythmic bursting to tonic spiking (a boundary between yellow and black in *Figure 4A*) looks like a hyperbola ($\bar{g}_{CAN} \times synaptic\ strength = const$).

We further investigated and compared the effect of reducing $\bar{g}_{CAN}$ or the synaptic strength on network activity amplitude and frequency as well as the effects on the recruitment of neurons not involved in rhythm generation (*Figure 4C–F*). To make this comparison, we picked a starting point in the 2D parameter space of $\bar{g}_{CAN}$ and synaptic strength where the network is bursting. Then in separate simulations, we gradually reduced either $\bar{g}_{CAN}$ or the synaptic strength to zero. We show that reducing either $\bar{g}_{CAN}$ or the synaptic strength have very similar effects on network activity amplitude and frequency (*Figure 4C,D*). Furthermore, de-recruitment of neurons in both cases is nearly identical (*Figure 4E,F*). Reducing either $\bar{g}_{CAN}$ or the synaptic strength decreases the excitatory input to the neurons during network oscillations which is a major component of the inspiratory drive potential. Therefore, in the $Ca_{Syn}$ network, manipulations of $\bar{g}_{CAN}$ will affect the strength of the inspiratory drive potential in the rhythmic inspiratory neurons in a way that is equivalent to changing the synaptic

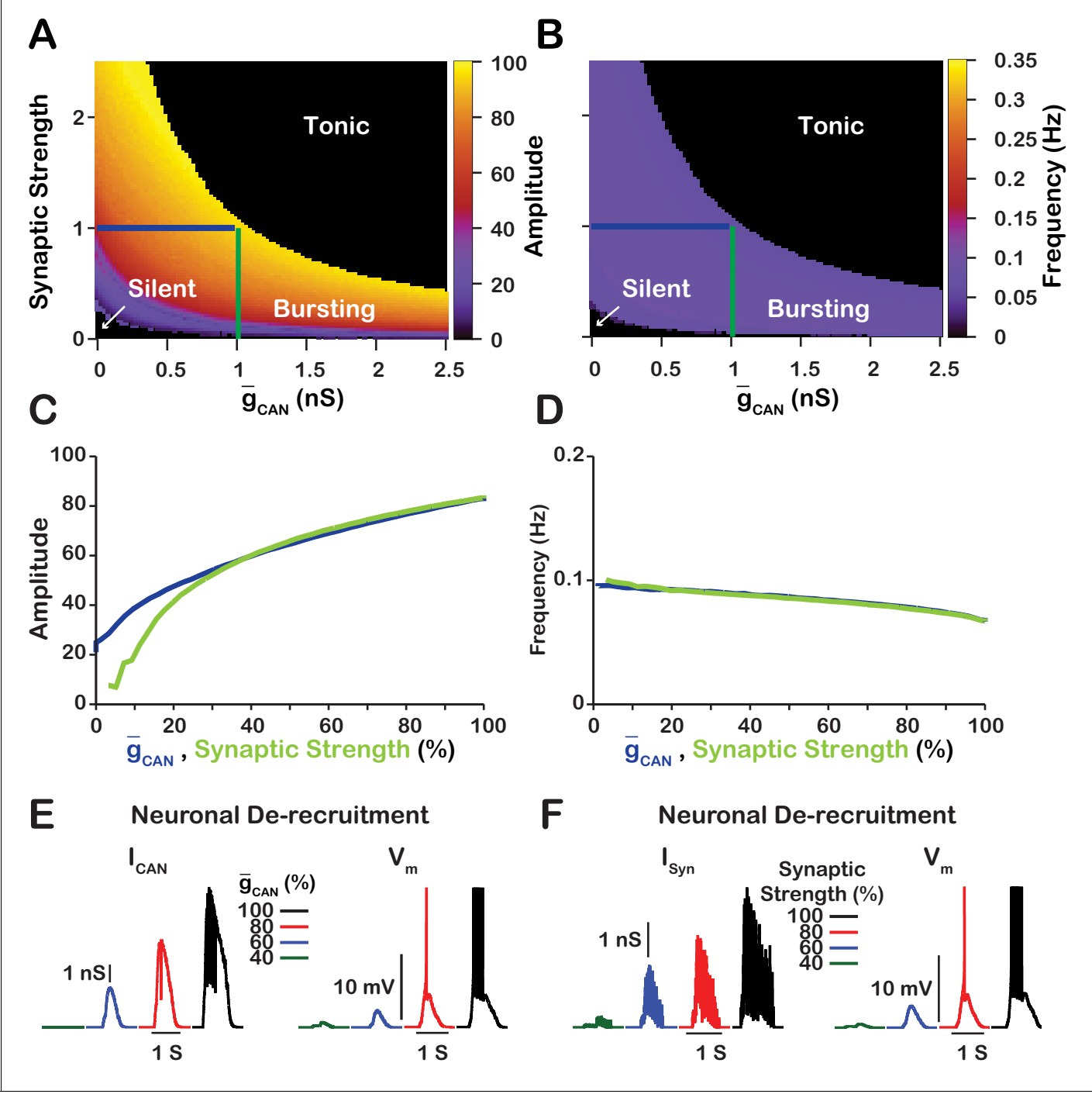

**Figure 4.** Manipulations of synaptic strength ($N \cdot P_{Syn} \cdot \frac{1}{2} W_{max}$) and $\bar{g}_{CAN}$ have equivalent effects on network activity amplitude, frequency and recruitment of inspiratory neurons not involved in rhythm generation. (A and B) Relationship between $\bar{g}_{CAN}$ (mean values for the simulated populations), synaptic strength and the amplitude and frequency in the $Ca_{Syn}$ network. Notice the symmetry about the X=Y line in panels A and B, which, indicates that changes in $\bar{g}_{CAN}$ and or synaptic strength are qualitatively equivalent. Synaptic strength was changed by varying $W_{max}$. (C) Relationship between network activity amplitude and the reduction of $\bar{g}_{CAN}$ (blue) or synaptic strength (green). (D) Relationship between network frequency and the reduction of $\bar{g}_{CAN}$ (blue) or synaptic strength (green). (E and F) Decreasing $\bar{g}_{CAN}$ or synaptic strength de-recruits neurons by reducing the inspiratory drive potential, indicated by the amplitude of subthreshold depolarization, right traces. The solid blue and green lines in panels A and B represent the location in the 2D parameter space of the corresponding blue and green curves in C and D. The action potentials in the right traces of E and F are

*Figure 4 continued on next page*

*Figure 4 continued*

truncated to show the change in neuronal inspiratory drive potential. The parameters used for these simulations are $\textbf{\textit{Ca}}_{\textbf{\textit{Syn}}}$: $\bar{\textbf{\textit{g}}}_{\textbf{\textit{Ca}}} = [0, 0]$, $\textbf{\textit{P}}_{\textbf{\textit{Ca}}} = 0.01$, $\textbf{\textit{P}}_{\textbf{\textit{Syn}}} = 1.0$ and $\textbf{\textit{W}}_{\textbf{\textit{max}}} = \textbf{\textit{var}}$.

DOI: https://doi.org/10.7554/eLife.41555.005

strength of the network. In contrast, manipulations of $\bar{g}_{CAN}$ in the $Ca_V$ network will only slightly affect the inspiratory drive potential due to changes in the average firing rate of active neurons (see *Figure 3B*).

## Robustness of amplitude and frequency effects

We also examined if the effects are conserved in both the $Ca_V$ and $Ca_{Syn}$ networks over a range of network parameters. To test this, we investigated the dependence of network activity amplitude and frequency on $\bar{g}_{CAN}$ and average synaptic strength for $Ca_{Syn}$ and $Ca_V$ networks with high ($P_{Syn} = 1$) and low ($P_{Syn} = 0.05$) connection probabilities, and high ($g_{Ca} = 0.1 \ nS$, $P_{Ca} = 0.1$), medium ($g_{Ca} = 0.01 \ nS$, $P_{Ca} = 0.01$) and low ($g_{Ca} = 0.001 \ nS$, $P_{Ca} = 0.005$) strengths of calcium sources (*Figure 5* and *6*). We found that changing the synaptic connection probability and changing the strength

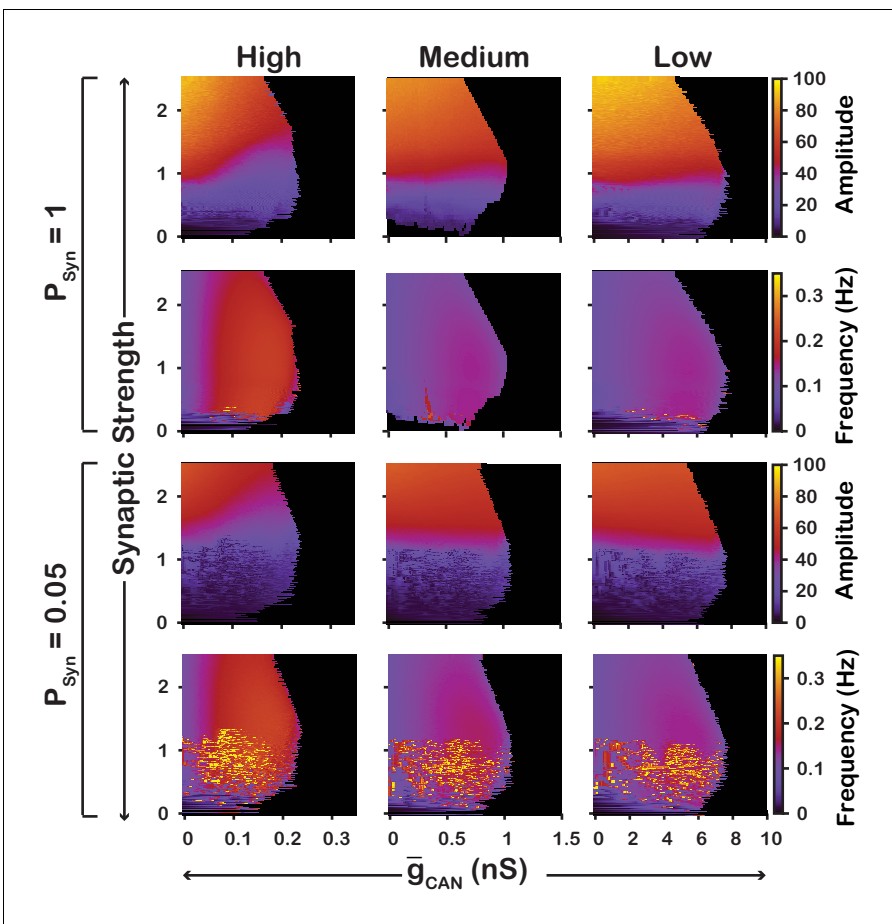

**Figure 5.** Robustness of amplitude and frequency effects to changes in $\bar{g}_{CAN}$ and synaptic strength in the $\textbf{\textit{Ca}}_{\textbf{\textit{V}}}$ network for 'high' (left), 'medium' (middle) and 'low' (right) conductance of the voltage-gated calcium channel $\textbf{\textit{I}}_{\textbf{\textit{Ca}}}$ as well as 'high' (top) and 'low' (bottom) network connection probabilities. Amplitude and frequency are indicated by color (scale bar at right). Black regions indicate tonic network activity. Values of $\bar{g}_{CAN}$ indicated are the mean values for the simulated neuronal populations.

DOI: https://doi.org/10.7554/eLife.41555.006

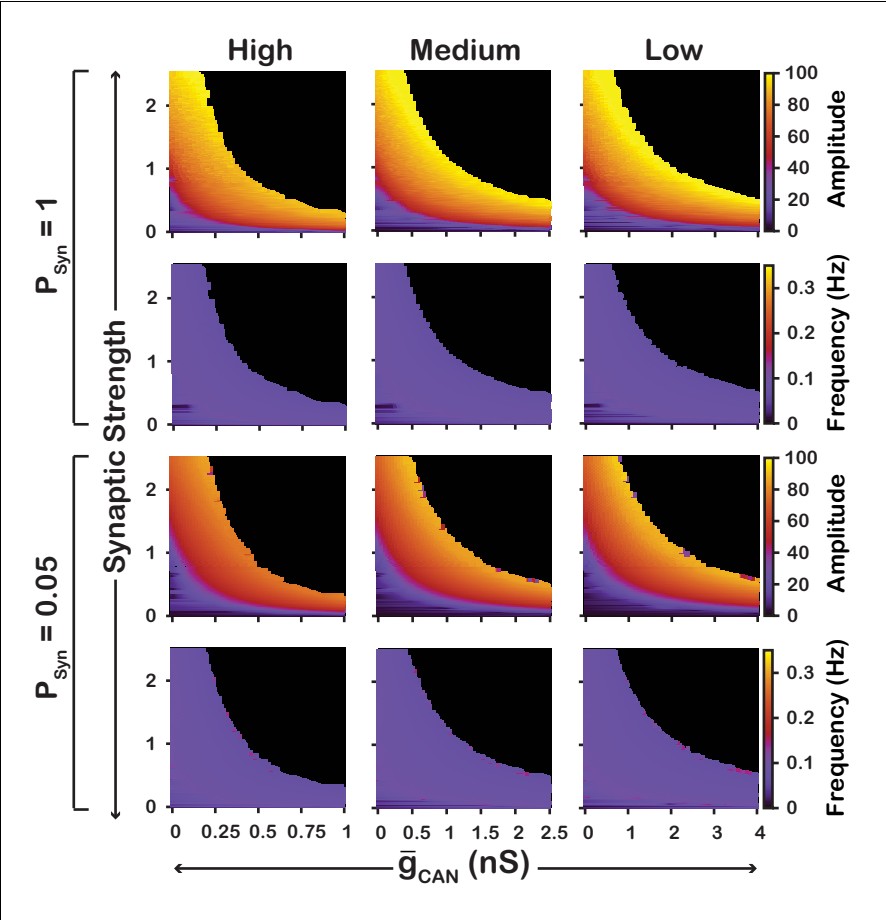

**Figure 6.** Robustness of amplitude and frequency effects to changes in $\bar{g}_{CAN}$ and synaptic strength in the $Ca_{Syn}$ network for 'high' (left), 'medium' (middle) and 'low' (right) calcium conductance in synaptic currents as well as 'high' (top) and 'low' (bottom) network connection probabilities. Amplitude and frequency are indicated by color (scale bar at right). Black regions indicate tonic network activity. Values of $\bar{g}_{CAN}$ indicated are the mean values for the simulated neuronal populations.

DOI: https://doi.org/10.7554/eLife.41555.007

of the calcium sources has no effect on the general relationship between $\bar{g}_{CAN}$ and the amplitude or frequency of bursts in the $Ca_V$ or $Ca_{Syn}$ networks. In other words, the general effect of increasing $\bar{g}_{CAN}$ on amplitude and frequency is conserved in both networks regardless of the synaptic connection probability or strength of the calcium sources. Increasing the strength of the calcium sources does, however, affects the range of possible $\bar{g}_{CAN}$ values where both networks produce rhythmic activity.

To summarize, in the $Ca_V$ model, increasing $\bar{g}_{CAN}$ increases frequency, through increased excitability but has no effect on amplitude. In contrast, in the $Ca_{Syn}$ model, increasing $\bar{g}_{CAN}$ slightly decreases frequency and increases amplitude. In this case, increasing $\bar{g}_{CAN}$ acts as a mechanism to increase the inspiratory drive potential and recruit previously silent neurons. Additionally, these features of the $Ca_V$ and $Ca_{Syn}$ models are robust and conserved across a wide range of network parameters.

## Intracellular calcium transients activating $I_{CAN}$ primarily result from synaptically activated sources

In experiments where $I_{CAN}$/TRPM4 was blocked by bath application of FFA or 9-phenanthrol (*Koizumi et al., 2018*) in vitro, the amplitude of network oscillations was strongly reduced and their frequency remained unchanged or was reduced insignificantly. Our model revealed that the effects

of $I_{CAN}$ blockade on amplitude and frequency depend on the source(s) of intracellular calcium (see *Figures 1* and *2*). If the calcium influx is exclusively voltage-gated, our model predicts that $I_{CAN}$ blockade will have no effect on amplitude but reduce the frequency. In contrast, if the calcium source is exclusively synaptically gated, our model predicts that blocking $I_{CAN}$ will strongly reduce the amplitude and slightly increase the frequency. Therefore, a multi-fold decrease in amplitude, seen experimentally, is consistent with the synaptically driven calcium influx mechanisms, while nearly constant bursting frequency may be due to calcium influx through both voltage- and synaptically gated channels. Following the predictions above, to reproduce experimental data, we incorporated both mechanisms in the full model and inferred their individual contributions by finding the best fit. We found that the best match is observed (*Figure 7*) if synaptically mediated and voltage-gated calcium influxes comprise about 95% and 5% of the total calcium influx, respectively. We note that some experiments have shown a small (~20%) reduction of inspiratory burst frequency accompanying larger reductions in pre-BötC population activity amplitude with $I_{CAN}$ block (*Peña et al., 2004*). In our model, such perturbations of frequency can occur if the contribution of voltage-gated calcium influxes is larger than indicated above, or if neuronal background calcium concentration which partially activates $I_{CAN}$ is higher than specified in the model.

### $I_{NaP}$-dependent and $[Ca]_i$-$I_{CAN}$-sensitive intrinsic bursting

In our model, we included $I_{NaP}$, $I_{CAN}$ as well as voltage-gated and synaptic mechanisms of $Ca^{2+}$ influx. Activation of $I_{CAN}$ by $Ca_{Syn}$ is the equivalent mechanism used in computational group-pacemaker

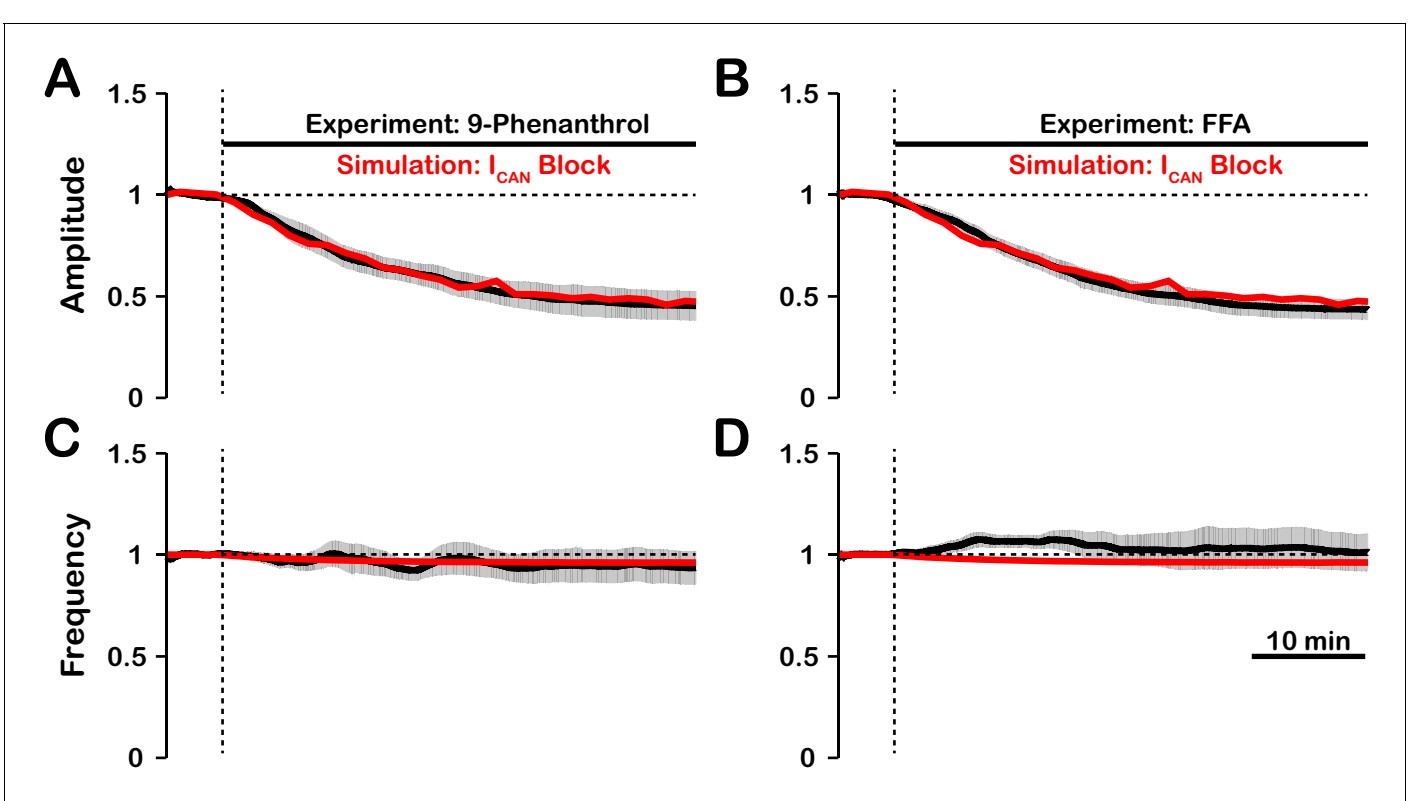

**Figure 7.** Experimental and simulated pharmacological blockade of $I_{CAN}$ by (A and C) 9-phenanthrol and (B and D) flufenamic acid (FFA). Both voltage-gated and synaptic sources of intracellular calcium are included. Experimental blockade of $I_{CAN}$ (black) by 9-phenanthrol and FFA significantly reduce the (A and B) amplitude of network oscillations while having little effect on (C and D) frequency. The black line represents the mean and the gray band is the S.E.M. of experimental integrated XII output recorded from neonatal rat brainstem slices in vitro, reproduced from *Koizumi et al., 2018*. Simulated blockade of $I_{CAN}$ (red) closely matches the reduction in (A and B) amplitude of network oscillations and slight decrease in (C and D) frequency seen with 9-phenanthrol and FFA. Simulated and experimental blockade begins at the vertical dashed line. Blockade was simulated by exponential decay of $\bar{g}_{CAN}$ with the following parameters: 9-phenanthrol: $\gamma_{Block} = 0.85$, $\tau_{Block} = 357s$; FFA: $\gamma_{Block} = 0.92$, $\tau_{Block} = 415s$. The network parameters are: $\bar{g}_{Ca} = 0.00175$ (*nS*), $P_{Ca} = 0.0275$, $P_{Syn} = 0.05$ and $W_{max} = 0.096$ (*nS*).
DOI: https://doi.org/10.7554/eLife.41555.008

models (*Rubin et al., 2009a*; *Song et al., 2015*). Rhythmic burst generation and termination in our model, however, are dependent on $I_{NaP}$ (*Butera et al., 1999a*). We investigated the sensitivity of intrinsic bursting in our model to $I_{NaP}$ and calcium channel blockade (*Figure 8*). Intrinsic bursting was identified in neurons by zeroing the synaptic weights to simulate synaptic blockade. $I_{NaP}$ and $I_{Ca}$ blockade was simulated by setting $\bar{g}_{NaP}$ and $\bar{g}_{Ca}$ to $0\,nS$. We found that after decoupling the network ($W_{max} = 0$) a subset of neurons remained rhythmically active (7% in this simulation) and that these were all neurons with a high $I_{NaP}$ conductance. In these rhythmically active neurons, bursting was abolished in all neurons by $I_{NaP}$ blockade. Interestingly, $I_{Ca}$ blockade applied before $I_{NaP}$ block

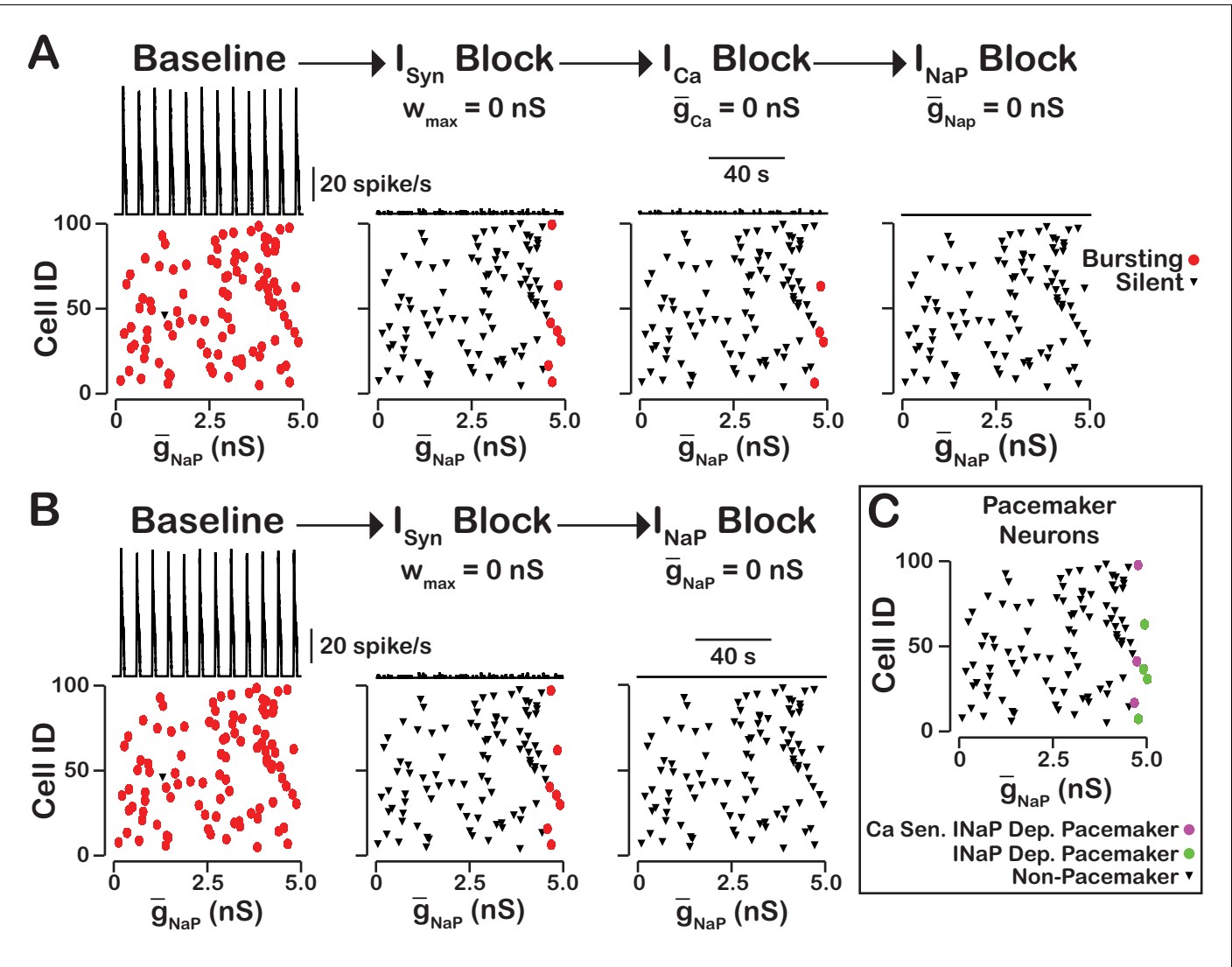

**Figure 8.** $I_{NaP}$-dependent and $Ca^{2+}$-sensitive intrinsic bursting. (A) From left to right, intrinsic bursters (pacemakers) are first identified by blocking synaptic connections. Cells whose activity is elmiated under these conditions are non-pacemaker neurons. Then, calcium sensitive neurons are silenced and identified by $I_{Ca}$ blockade. The remaining neurons are identifed as sensitive to $I_{NaP}$ block. Top traces show the network output and Cell ID vs. $\bar{g}_{NaP}$ scatter plots that identify silent and bursting neurons under each condition. (B) $I_{NaP}$ blockade after synaptic blockade eliminates bursting in all neurons. Therefore, all intrinsic bursters are $I_{NaP}$ dependent. (C) Identification of calcium-sensitive and $I_{NaP}$-dependent as well as calcium-insensitive and $I_{NaP}$-dependent intrinsic bursters. Notice that only the neurons with the highest value of $\bar{g}_{NaP}$ are intrinsic bursters and that a subset of these neurons are sensitive to calcium blockade but all are dependent on $I_{NaP}$. The network parameters are: $\bar{g}_{Ca} = 0.00175$ $(nS)$, $P_{Ca} = 0.0275$, $P_{Syn} = 0.05$ and $W_{max} = 0.096$ $(nS)$. The values of $\bar{g}_{NaP}$ given in the scatter plots indicate the magnitude of $\bar{g}_{NaP}$ for each neuron in the network and show the range of the $\bar{g}_{NaP}$ distribution.

DOI: https://doi.org/10.7554/eLife.41555.009

abolished intrinsic bursting in three of the seven neurons and $I_{NaP}$ block applied afterwards abolished intrinsic bursting in the remaining four neurons. Although only one rhythmogenic ($I_{NaP}$-based) mechanism exists in this model, bursting in a subset of these intrinsically bursting neurons is calcium sensitive, consistent with experimental observations of calcium-sensitive intrinsic bursters (*Thoby-Brisson and Ramirez, 2001*; *Del Negro et al., 2005*; *Peña et al., 2004*). In calcium-sensitive bursters, $Ca^{2+}$ blockade in our model abolishes bursting by reducing the intracellular calcium concentration and, hence, $I_{CAN}$ activation, which ultimately reduces excitability. We note that in our model the numbers of $I_{NaP}$-dependent and calcium-sensitive intrinsic bursters will vary depending on the mean and width of the $I_{NaP}$ distribution and the background intracellular calcium concentration.

## The rhythmogenic kernel

Our simulations have shown that the primary role of $I_{CAN}$ is amplitude but not oscillation frequency modulation with little or no effect on network activity frequency. Here we examined the neurons that remain active and maintain rhythm after $I_{CAN}$ blockade (*Figure 9*). We found that the neurons that remain active are primarily neurons with the highest $\bar{g}_{NaP}$ and that bursting in these neurons is dependent on $I_{NaP}$. Some variability exists and neurons with relatively low $\bar{g}_{NaP}$ value can remain active due to synaptic interactions while a neuron with a slightly higher $\bar{g}_{NaP}$ without sufficient synaptic input may become silent. These neurons that remain active after compete blockade of $I_{CAN}$ form a $I_{NaP}$-dependent kernel of a rhythm generating circuit.

## Intracellular calcium transients and network activity during inhibition of $I_{CAN}$/TRPM4

Dynamic calcium imaging has been used to assess activity of the population of pre-BöC excitatory neurons as well as individual pre-BötC neurons during pharmacological inhibition of $I_{CAN}$/TRPM4 in vitro (*Koizumi et al., 2018*). These experiments indicate that network output activity amplitude and pre-BötC excitatory population-level intracellular calcium transients are highly correlated while the network oscillation frequency is not significantly perturbed. Interestingly, during $I_{CAN}$/TRPM4 block, changes in calcium transients of individual neurons can differ significantly from the average population-level calcium transients. To assess if our model is consistent with these experimental results and

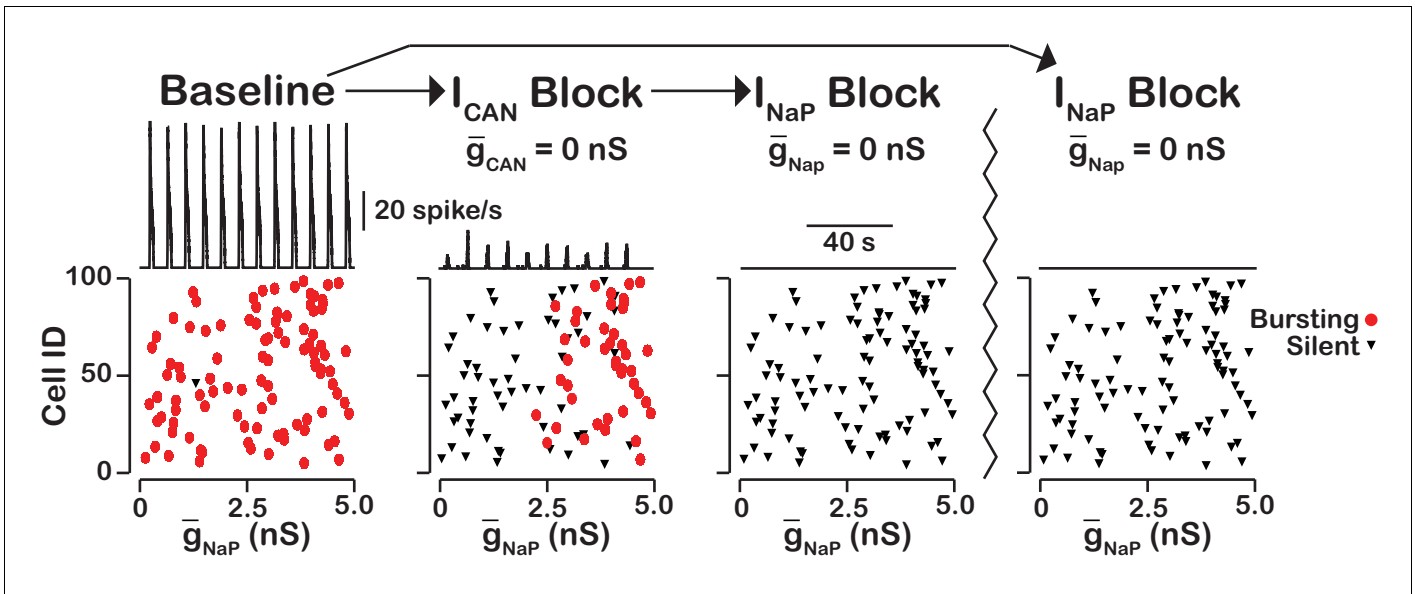

**Figure 9.** $I_{CAN}$ blockade reveals an $I_{NaP}$-dependent rhythmogenic kernel. The top traces show the network output at baseline, after $I_{CAN}$ blockade and $I_{NaP}$ blockade. The bottom Cell ID vs. $\bar{g}_{NaP}$ scatter plots identify silent and bursting neurons in each conditon. Notice that only neurons with relitively high $\bar{g}_{NaP}$ remain active after $I_{CAN}$ block. The network parameters used are: $\bar{g}_{Ca} = 0.00175$ (*nS*), $P_{Ca} = 0.0275$, $P_{Syn} = 0.05$ and $W_{max} = 0.096$ (*nS*). The values of $\bar{g}_{NaP}$ given in the scatter plots indicate the magnitude of $\bar{g}_{NaP}$ for each neuron in the network and show the range of the $\bar{g}_{NaP}$ distribution.
DOI: https://doi.org/10.7554/eLife.41555.010

gain additional insight into intracellular calcium dynamics during network activity, we analyzed simultaneous changes in the amplitude of network neuronal spiking activity, the average intracellular calcium concentration ($[Ca]_i$) of all network neurons, as well as $[Ca]_i$ of individual neurons, with different network connection probabilities ($P_{Syn}$) during simulated $I_{CAN}$ block (*Figure 10*). We found that regardless of $P_{Syn}$, the network activity amplitude and average intracellular calcium concentration are highly correlated (*Figure 10A,B*). $P_{Syn}$ has no effect on the relationship between amplitude, calcium transients at the network level, or network oscillation frequency provided that the synaptic strength remains constant ($N \cdot P_{Syn} \cdot \frac{1}{2}W_{max} = const$). $P_{Syn}$ does, however, affect the change in the peak $[Ca]_i$ in individual neurons. In a network with a high connection probability ($P_{Syn} = 1$) the synaptic current/calcium transient is nearly identical for all neurons and therefore the change in $[Ca]_i$ during $I_{CAN}$ blockade is approximately the same for each neuron (*Figure 10C*). In a sparsely connected network, as proposed for the connectivity of the pre-BötC network (*Carroll et al., 2013*; *Carroll and Ramirez, 2013*) the synaptic current and calcium influx are more variable and reflect the heterogeneity in spiking frequency of the pre-synaptic neurons (*Figure 10*). Interestingly, in a network with low connection probability ($P_{Syn}$<0.1), the peak $[Ca]_i$ transient in some neurons increases when $I_{CAN}$ is blocked (*Figure 10E*), consistent with our experimental results (*Koizumi et al., 2018*).

## Discussion

Establishing cellular and circuit mechanisms generating the rhythm and amplitude of respiratory oscillations in the mammalian brainstem pre-BötC has remained an unsolved problem of widespread interest in neurophysiology since this structure, essential for breathing to support mammalian life, was discovered nearly three decades ago (*Smith et al., 1991*). Our objective in this theoretical study was to re-examine and further define contributions of two of the main currently proposed neuronal biophysical mechanisms operating in pre-BötC excitatory circuits, specifically mechanisms involving $I_{CAN}$ activated by neuronal calcium fluxes (*Thoby-Brisson and Ramirez, 2001*; *Peña et al., 2004*; *Del Negro et al., 2005*; *Mironov, 2008*; *Rubin et al., 2009a* ) and voltage-dependent $I_{NaP}$ in the circuit neurons (*Butera et al., 1999a*; *Del Negro et al., 2002*; *Koizumi and Smith, 2008*; *Koizumi and Smith, 2008*). While these sodium- and calcium-based mechanisms have been studied extensively over the past two-decades and shown experimentally to be integrated in pre-BötC circuits, their actual roles in circuit operation are continuously debated and unresolved (*Rybak et al., 2014*). Both mechanisms have been proposed to be fundamentally involved in rhythm generation either separately or in combination, as plausibly shown from previous theoretical modeling studies (*Butera et al., 1999a*; *Jasinski et al., 2013*; *Rubin et al., 2009a*; *Toporikova and Butera, 2011*). Furthermore, the process of rhythm generation in pre-BötC circuits must be associated with an amplitude of excitatory circuit activity sufficient to drive downstream circuits to produce adequate respiratory motor output. Biophysical mechanisms involved in generating excitatory population activity amplitude have also not been established. Our analysis is motivated by the experimental observations obtained from neonatal rodent slices isolating pre-BötC circuits in vitro that inhibition of the endogenously active $I_{CAN}$/TRPM4 strongly reduces the amplitude of network oscillations within pre-BötC circuits but has a small effect on oscillation frequency (*Peña et al., 2004*; *Koizumi et al., 2018*). These findings challenge currently proposed $I_{CAN}$-based models for rhythm generation in the isolated pre-BötC and indicate a functional organization of pre-BötC circuits, in terms of oscillatory frequency and amplitude generation, that needs to be defined.

We accordingly analyzed the role of $I_{CAN}$ and possible sources of intracellular calcium transients activating this conductance and found that the effect of simulated $I_{CAN}$ blockade on amplitude and frequency is highly dependent on the source(s) of intracellular calcium, which is also a central issue to be resolved. In the case where $Ca_{Syn}$ is the primary intracellular calcium source, $I_{CAN}$ blockade generates a large reduction in network activity amplitude. In contrast, when $Ca_V$ is the only intracellular calcium source, $I_{CAN}$ blockade has little effect on network activity amplitude and primarily affects the population bursting frequency that is caused by decreased excitability. Additionally, we show that activation of $I_{CAN}$ by $Ca_{Syn}$ functions as a mechanism to augment the inspiratory drive potential and amplitude of population activity, and that this effect is similar to increasing the synaptic coupling strength within the network. Therefore, in the case of $Ca_{Syn}$, blockade of $I_{CAN}$ reduces the inspiratory drive potential causing de-recruitment of non-pacemaker rhythmic neurons and reduction of network

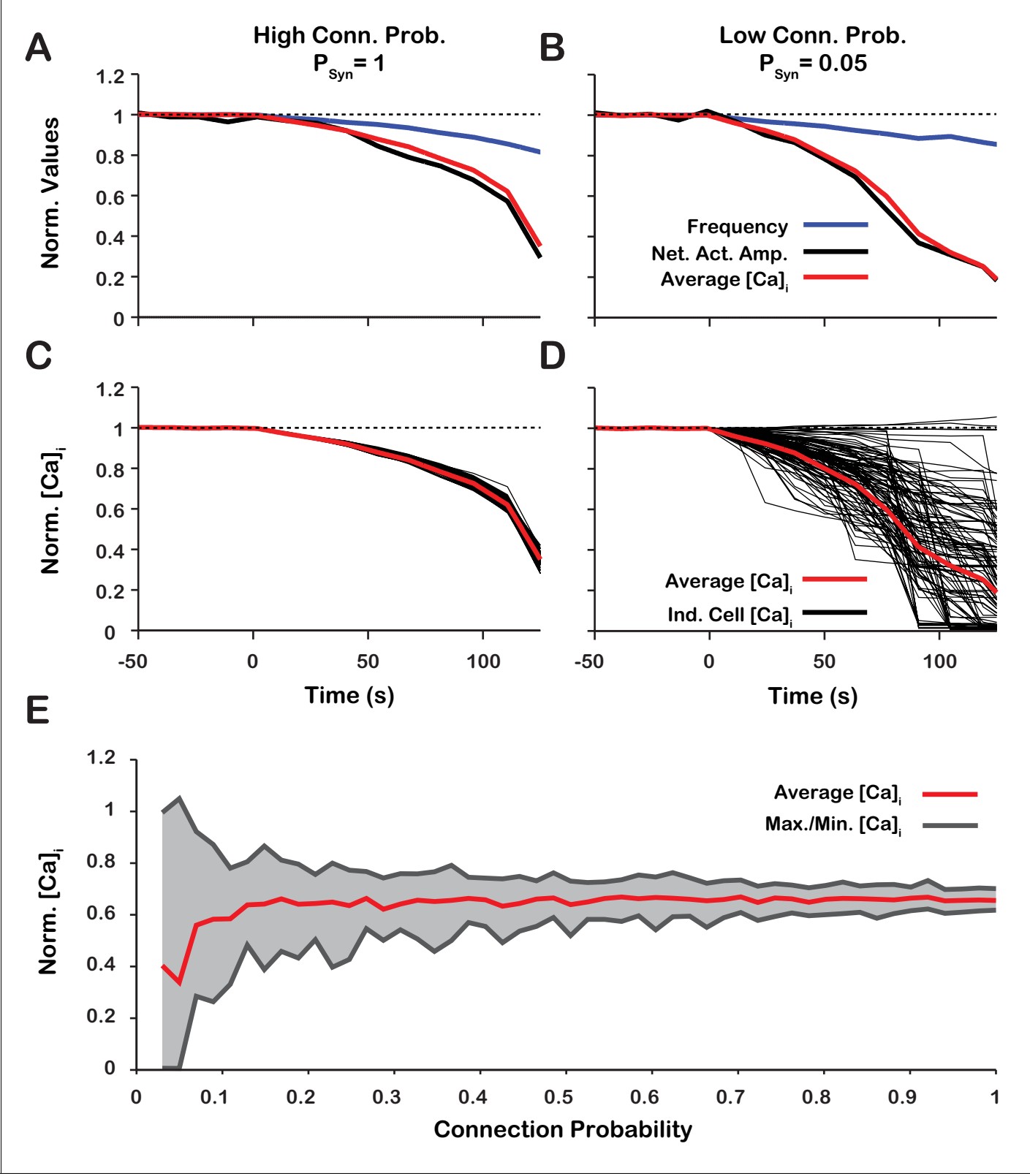

**Figure 10.** Changes of network activity amplitude, average network intracellular calcium concentration $[Ca]_i$ amplitude, and single model neuron $[Ca]_i$ amplitude during simulated $I_{CAN}$ blockade. (A and B) Effect of $I_{CAN}$ block on network activity amplitude, network calcium amplitude and frequency for network connection probabilities (A) P = 1 and (B) P = 0.05. (C and D) Effect of $I_{CAN}$ block on changes in the magnitude of peak cellular calcium transients for network connection probabilities (C) $P_{Syn} = 1$ and (D) $P_{Syn} = 0.05$. (E) Maximum, minimum and average change in the peak intracellular

*Figure 10 continued on next page*

*Figure 10 continued*

calcium transient of individual neurons as a function of synaptic connection probability. All curves in A through E are normalized to their baseline values. Synaptic weight was adjusted to keep the average synaptic strength ($N \cdot P_{Syn} \cdot \frac{1}{2} W_{max} = const$) constant. Notice that lowering the synaptic connection probability increases the variability in the peak intracellular calcium concentration during $I_{CAN}$ blockade. Interestingly, for connection probabilities below approximately 5%, blocking $\bar{g}_{CAN}$ can increase the peak calcium transient in a small subset of neurons. The network parameters used are: $\bar{g}_{Ca} = 0.00175$ ($nS$) and $P_{Ca} = 0.0275$ and $W_{max} = var$.

DOI: https://doi.org/10.7554/eLife.41555.011

activity amplitude. In a model where $I_{CAN}$ is activated by both $Ca_V$ and $Ca_{Syn}$ with contributions of 5% and 95% respectively, we show that simulated blockade of $I_{CAN}$ generates a large reduction in network population activity amplitude and a slight decrease in frequency. This closely reproduces experimental blockade of $I_{CAN}$/TRPM4 by either 9-phenanthrol or FFA (*Figure 7*). Finally, we showed that the change in the peak calcium transients for individual neurons during $I_{CAN}$ blockade, particularly at relatively low network connection probabilities ($P_{Syn} \sim <0.1$), are consistent with experimental data.

## Role of $I_{CAN}$ in the pre-BötC respiratory network

The hypothesis that $I_{CAN}$ is involved in generation of the inspiratory rhythm is based on experimental observations from in vitro mouse medullary slice preparations (*Pace et al., 2007*; *Mironov, 2008*; *Del Negro et al., 2005*; *Del Negro et al., 2010*; *Peña et al., 2004*; *Thoby-Brisson and Ramirez, 2001*), and in silico modeling studies (*Jasinski et al., 2013*; *Rybak et al., 2003*; *Toporikova and Butera, 2011*). Theories of $I_{CAN}$-dependent bursting rely on intracellular $Ca^{2+}$ signaling mechanisms that have not been well defined.

Two models of $I_{CAN}$-dependent rhythmic bursting in vitro have been proposed and are referred to as the 'dual pacemaker' and 'group pacemaker' models. In the dual pacemaker model, two types of pacemaker neurons are proposed that are either $I_{NaP}$-dependent (riluzole sensitive) or $I_{CAN}$-dependent ($Cd^{2+}$ sensitive) intrinsic bursters (inspiratory 'pacemaker' neurons) (see *Rybak et al., 2014* for review). In this model network, oscillations are thought to originate from these pacemaker neurons which through excitatory synaptic interactions synchronize bursting and drive activity of other rhythmic inspiratory neurons within the pre-BötC. Although pacemaker neurons sensitive to neuronal $Ca^{2+}$ flux blockade through $Ca_V$ have been reported (*Peña et al., 2004*; *Thoby-Brisson and Ramirez, 2001*, the source and mechanism driving intracellular $Ca^{2+}$ oscillations has not been definitely delineated. Computational models of $I_{CAN}$-dependent pacemaker neurons rely on mechanisms for burst initiation and termination, for example IP3-dependent $Ca^{2+}$ oscillations (*Toporikova and Butera, 2011*; *Del Negro et al., 2010*), that have been questioned from recent negative experimental results (*Beltran-Parrazal et al., 2012*; *Toporikova et al., 2015*). In favor of the dual pacemaker concept, experimental evidence has been presented that pharmacological block of both $I_{CAN}$ and $I_{NaP}$ are necessary to disrupt rhythmogenesis under normoxic conditions in vitro (*Peña et al., 2004*). Other experimental results suggest that blocking $I_{NaP}$ alone in vitro is sufficient (*Koizumi and Smith, 2008*; *Toporikova et al., 2015*), and we have discussed from a theoretical perspective how such discrepancies may depend on network excitability state and connectivity (*Jasinski et al., 2013*). Our present model defines more clearly parameters such as calcium flux sources and background calcium levels that can influence the relative contributions of $I_{CAN}$ in various network states to be further explored in model simulation studies.

In versions of the 'group pacemaker' model (*Rubin et al., 2009a*; *Del Negro et al., 2010*; *Feldman and Del Negro, 2006*), network oscillations are initiated through recurrent synaptic excitation that trigger postsynaptic $Ca^{2+}$ influx. Subsequent $I_{CAN}$ activation generates membrane depolarization (inspiratory drive potential) to drive neuronal bursting. Synaptically triggered $Ca^{2+}$ influx and the contribution of $I_{CAN}$ to the inspiratory drive potential of individual pre-BötC neurons are experimentally supported (*Mironov, 2008*; *Pace et al., 2007*); however, the mechanism of burst termination remains unclear. Again, the computational group-pacemaker models that have been explored (*Rubin et al., 2009a*) rely on speculative mechanisms for burst termination that need to be tested experimentally, and in some cases lack key biophysical features of the pre-BötC neurons such as voltage-dependent frequency control and expression of $I_{NaP}$.

In our model, we showed that blockade of either $I_{CAN}$ or synaptic interactions produce qualitatively equivalent effects on network population activity amplitude and frequency when the calcium transients are primarily generated from synaptic sources (*Figure 4*). Consequently, our model predicts that blockade of $I_{CAN}$ or synaptic interactions in the isolated pre-BötC in vitro will produce comparable effects on amplitude and frequency. This is the case as *Johnson et al. (1994)* showed that gradual blockade of synaptic interactions by low calcium solution significantly decreases network activity amplitude while having little effect on frequency, similar to the experiments where the $I_{CAN}$ channel TRPM4 is blocked with 9-phenanthrol (*Koizumi et al., 2018*). We note that complete blockade of $I_{CAN}$ in our model can ultimately abolish synchronized network oscillations due to weakened excitatory synaptic transmission, which results in neuronal de-recruitment and desynchronization of the network, particularly when synaptic strength is low (*Figure 4*). Thus, $I_{CAN}$ plays a critical role in network activity synchronization that determines the ability of the pre-BötC excitatory network to produce rhythmic output, depending on synaptic strength.

Overall, our new model simulations for the isolated pre-BötC excitatory network suggest that the role of $I_{CAN}$/TRPM4 activation is to amplify excitatory synaptic drive in generating the amplitude of inspiratory population activity, essentially independent of the biophysical mechanism generating inspiratory rhythm. We note that the recent experiments have also shown that in the more intact brainstem respiratory network that ordinarily generates patterns of inspiratory and expiratory activity, endogenous activation of $I_{CAN}$/TRPM4 appears to augment the amplitude of both inspiratory and expiratory population activity, and hence these channels are fundamentally involved in inspiratory-expiratory pattern formation (Koizumi et al., 2018).

## Intracellular calcium dynamics and network activity during inhibition of $I_{CAN}$/TRPM4

We analyzed the correlation between calcium transients and inspiratory activity of individual inspiratory neurons as well as the entire network, particularly since dynamic calcium imaging has been utilized to assess activity of individual cells and populations of pre-BötC excitatory neurons in vitro during pharmacological inhibition of $I_{CAN}$/TRPM4 (*Koizumi et al., 2018*). We show that intracellular calcium transients are highly correlated with network and cellular activity across the duration of an $I_{CAN}$ blockade simulation, consistent with experimental observations.

Additionally, we examined the relative change in the peak calcium transients in single neurons as a function of $I_{CAN}$ conductance. We show that in a subset of neurons the peak calcium transient increases with reduced $I_{CAN}$. This result is surprising but is also supported by the calcium imaging data (*Koizumi et al., 2018*). This occurs in neurons that receive most of their synaptic input from pacemaker neurons and our analyses suggest this is possible in sparse networks, that is with low connection probability. In pacemaker neurons, $I_{CAN}$ blockade leads to a reduction of their excitability resulting in an increased value of $I_{NaP}$ inactivation gating variable at the burst onset. Thus, during the burst, the peak action potential frequency and the synaptic output from these neurons is increased with $I_{CAN}$ blockade. Consequently, neurons that receive synaptic input from pacemaker neurons will see an increase in their peak calcium transients. In most neurons, however, synaptic input is received primarily from non-pacemaker rhythmic neurons. Since $I_{CAN}$ blockade de-recruits these non-pacemaker neurons, the synaptic input and subsequent calcium influx in most of these cells decreases. Therefore, our model predicts that in a sparse network, which has been proposed for pre-BötC network connectivity (*Carroll et al., 2013*; *Carroll and Ramirez, 2013*), blocking $I_{CAN}$ results in very diverse responses at the cellular level with an overall tendency to reduce intracellular calcium transients such that the amplitude of these transients averaged over the entire population decreases during $I_{CAN}$ blockade, while their burst frequency is essentially unchanged, as found from the calcium imaging experiments (*Koizumi et al., 2018*).

## Synaptic calcium sources

Our model suggests that calcium transients in the pre-BötC are coupled to excitatory synaptic input, that is pre-synaptic glutamate release and binding to post-synaptic glutamate receptors triggers calcium entry. The specific mechanisms behind this process are unclear; however, this is likely dependent on specific types of ionotropic or metabotropic glutamate receptors.

There are three subtypes of ionotropic glutamate receptors, N-methyl-D-aspartate (NMDA), Kainate (KAR), and a-amino-3-hydroxy-5-methyl-4-isoxazolepropionic acid (AMPA), all of which are expressed in the pre-BötC (*Paarmann et al., 2000*) and have varying degrees of calcium permeability. NMDA and AMPA are unlikely candidates for direct involvement in synaptically mediated calcium influx in the pre-BötC. Pharmacological blockade of NMDA receptors does not consistently effect the amplitude or frequency of XII motor output (*Lieske and Ramirez, 2006a*; *Morgado-Valle and Feldman, 2007*; *Pace et al., 2007*) and AMPA receptors in the pre-BötC show high expression of the subunit GluR2, which renders the AMPA ion channel pore impermeable to $Ca^{2+}$ (*Paarmann et al., 2000*). It is possible, however, that AMPA mediated depolarization may trigger calcium influx indirectly through the voltage-gated calcium channel activation on the post-synaptic terminal. The latter may contribute to synaptically triggered calcium influx as blockade of P/Q-type (but not L- N-type) calcium channels reduces XII motor output driven from the pre-BötC in the in vitro mouse slice preparations from normal animals (*Lieske and Ramirez, 2006a*; *Koch et al., 2013*).

Calcium permeability through KAR receptors is dependent on subunit expression. The KAR subunit GluK3 is highly expressed in the pre-BötC (*Paarmann et al., 2000*) and is calcium permeable (*Perrais et al., 2009*) making it a possible candidate for synaptically mediated calcium entry. Furthermore, GluK3 is insensitive to tonic glutamate release and only activated by large glutamate transients (*Perrais et al., 2009*). Consequently, GluK3 may only be activated when receiving synaptic input from a bursting presynaptic neuron which would presumably generate large glutamate transients. The role of GluK3 in the pre-BötC has not been investigated.

Metabotropic glutamate receptors (mGluR) indirectly activate ion channels through G-protein mediated signaling cascades. Group 1 mGluRs which include mGluR1 and mGluR5 are typically located on post-synaptic terminals (*Shigemoto et al., 1997*) and activation of group 1 mGluRs is commonly associated with calcium influx through calcium permeable channels (*Berg et al., 2007*; *Endoh, 2004*; *Mironov, 2008*) and calcium release from intracellular calcium stores (*Pace et al., 2007*).

In the pre-BötC, mGluR1/5 are thought to contribute to calcium influx by triggering the release of calcium from intracellular stores (*Pace et al., 2007*) and/or the activation of the transient receptor potential C3 (TRPC3) channel (*Ben-Mabrouk and Tryba, 2010*). Blockade of mGluR1/5 reduces the inspiratory drive potential in pre-BötC neurons (*Pace et al., 2007*) without significant perturbation of inspiratory frequency (*Pace et al., 2007*; *Lieske and Ramirez, 2006a*), which is consistent with the effects of $I_{CAN}$/TRPM4 blockade (*Koizumi et al., 2018*). TRPC3 is a calcium permeable channel (*Thebault et al., 2005*) that is associated with calcium signaling (*Hartmann et al., 2011*), store-operated calcium entry (*Kwan et al., 2004*), and synaptic transmission (*Hartmann et al., 2011*). TRPC3 is activated by diacylglycerol (DAG) (*Clapham, 2003*), which is formed after synaptic activation of mGluR1/5. TRPC3, which is highly expressed in pre-BötC glutamatergic inspiratory neurons, is often co-expressed with TRPM4 (*Koizumi et al., 2018*) and was hypothesized to underlie $I_{CAN}$ activation in the pre-BötC (*Ben-Mabrouk and Tryba, 2010*) and other brain regions (*Amaral and Pozzo-Miller, 2007*; *Zitt et al., 1997*). Furthermore, TRPC3 and $I_{CAN}$ have been shown to underlie slow excitatory post-synaptic current (sEPSC) (*Hartmann et al., 2008*; *Hartmann et al., 2011*). This is consistent with our model since $I_{CAN}$ activation is dependent on synaptically triggered calcium entry, and the calcium dynamics are slower than the fast AMPA based current $I_{Syn}$. Therefore, in our model, $I_{CAN}$ decays relatively slowly and, hence, can be treated as a sEPSC.

The effect of TRPC3 blockade by 3-pyrazole on pre-BötC network activity amplitude is remarkably similar to that with blockade of TRPM4 (*Koizumi et al., 2018*). This suggests that the $I_{CAN}$/TRPM4 activation may be dependent on/coupled to TRPC3. A possible explanation is that TRPC3, which is calcium permeable, mediates synaptically triggered calcium entry. It is also likely that TRPC3 plays a role in maintaining background calcium concentration levels. We tested this hypothesis by simulating the blockade of synaptically-triggered calcium influx while simultaneously lowering the background calcium concentration (*Figure 11*). These simulations generated large reductions in activity amplitude with no effect on frequency which are consistent with data from experiments where TRPC3 is blocked using 3-pyrazole (*Koizumi et al., 2018*). This indirectly suggests that TRPC3 may be critical for synaptically-triggered calcium entry and subsequent $I_{CAN}$ activation.

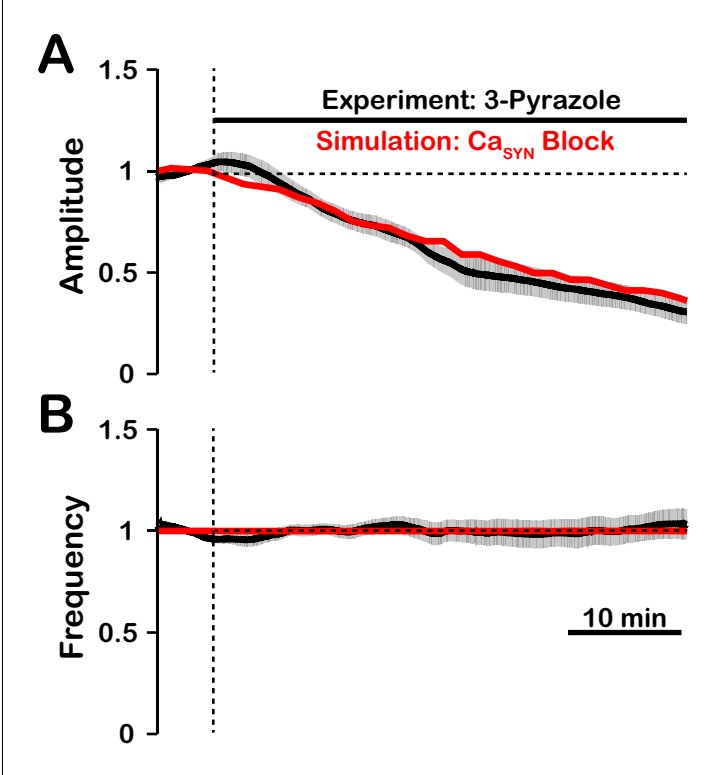

**Figure 11.** Comparison of experimental (black) and simulated (red) TRPC3 blockade (by $Ca_{Syn}$ block) on network activity amplitude (**A**) and frequency (**B**). Simulated and experimental blockade begins at the vertical dashed line. The black line represents the mean and the gray band represents the S.E.M. of experimental integrated XII output recorded from neonatal rat brainstem slices in vitro, reproduced from *Koizumi et al., 2018*. Blockade was simulated by exponential decay of $P_{Ca}$ with the following parameters: 3-pyrazole: $\gamma_{Block} = 1.0$, $\tau_{Block} = 522.5\,s$. The network parameters are: $\bar{g}_{Ca} = 0.00175$ (*nS*), $P_{Ca} = 0.0275$, $P_{Syn} = 0.05$ and $W_{max} = 0.096$ (*nS*).
DOI: https://doi.org/10.7554/eLife.41555.012

## $I_{NaP}$ -dependent rhythmogenic kernel

$I_{NaP}$ is a conductance present ubiquitously in pre-BötC inspiratory neurons, and is established to underlie intrinsic oscillatory neuronal bursting in the absence of excitatory synaptic interactions in neurons with sufficiently high $I_{NaP}$ conductance densities (*Del Negro et al., 2002*; *Koizumi and Smith, 2008*; *Koizumi and Smith, 2008*; *Yamanishi et al., 2018*). Accordingly, we randomly incorporated this conductance in our model excitatory neurons from a uniform statistical distribution to produce heterogeneity in $I_{NaP}$ conductance density across the population. Our simulations indicate that the circuit neurons mostly with relatively high $I_{NaP}$ conductance values underlie rhythm generation and remain active after compete blockade of $I_{CAN}$ in our model network, thus forming a $I_{NaP}$-dependent rhythmogenic kernel, including some neurons with intrinsic oscillatory bursting behavior when synaptically uncoupled.

As noted above, the rhythmogenic properties of individual neurons depend on whether their $I_{NaP}$ conductance is high enough and, therefore, the number of intrinsic bursters in the model is defined by the width of this conductance distribution over the population. However, the critical value of $I_{NaP}$ conductance and intrinsic bursting properties in general were also shown theoretically and experimentally to critically depend on the conductances of other ionic channels (e.g. leak and delayed rectifier potassium conductances) and extracellular ion concentrations such as potassium concentration (*Bacak et al., 2016b*; *Koizumi and Smith, 2008*; *Rybak et al., 2003*). Therefore, we believe that in reality the specific composition of the rhythmogenic kernel and its oscillatory capabilities strongly depend on the existing combination of neuronal conductances and on the in vitro experimental conditions.

Recently, it has become apparent that there is functional heterogeneity within pre-BötC excitatory circuits, including distinct subpopulations of neurons involved in generating periodic sighs (*Toporikova et al., 2015*; *Li et al., 2016*), arousal (*Yackle et al., 2017*), and the subpopulations generating regular inspiratory activity. Activity of the normal inspiratory and sigh-generating subpopulations in the pre-BötC isolated in vitro is proposed to be dependent on activation of $I_{NaP}$ (*Toporikova et al., 2015*). Our experimental and modeling results suggest that within the normal inspiratory population, there are subpopulations distinguished by their role in rhythm versus amplitude generation due to biophysical properties: there is a $I_{CAN}$/TRPM4-dependent recruitable population of excitatory neurons for burst amplitude generation and the $I_{NaP}$-dependent rhythmogenic kernel population. The spatial arrangements of these two synaptically interconnected excitatory populations within the pre-BötC are currently unknown, and it remains an important experimental problem to identify the cells constituting the rhythmogenic kernel and their biophysical properties. This should now be possible, since our analysis and experimental results suggest that the rhythmically active neurons of the kernel population can be revealed and studied after pharmacologically inhibiting the $I_{CAN}$/TRPM4-dependent inspiratory burst-generating population.

A 'burstlet theory' for emergent network rhythms has recently been proposed to account for inspiratory rhythm and pattern generation in the isolated pre-BötC in vitro (*Kam et al., 2013*; *Del Negro et al., 2018*). This theory postulates that a subpopulation of excitatory neurons generating small amplitude oscillations (burstlets) functions as the inspiratory rhythm generator that drives neurons that generate the larger amplitude, synchronized inspiratory population bursts. This concept emphasizes that subthreshold neuronal membrane oscillations need to be considered and that there is a neuronal subpopulation that functions to independently form the main inspiratory bursts. This is similar to our concept of distinct excitatory subpopulations generating the rhythm versus the amplitude of inspiratory oscillations. Biophysical mechanisms generating rhythmic burstlets and the large amplitude inspiratory population bursts in the burstlet theory are unknown, and the general problem of understanding the dynamic interplay of circuit interactions and cellular biophysical processes in the generation of population-level bursting activity has been highlighted (*Richter and Smith, 2014*; *Ramirez and Baertsch, 2018b*). We have identified a major $Ca^{2+}$-dependent conductance mechanism for inspiratory burst amplitude (pattern) generation and show theoretically how this mechanism may be coupled to excitatory synaptic interactions and is independent of the rhythm-generating mechanism. We also note that a basic property of $I_{NaP}$ is its ability to generate subthreshold oscillations and promote burst synchronization (*Butera et al., 1999b*; *Bacak et al., 2016a*). However, in contrast to our proposal for the mechanisms operating in the kernel rhythm-generating subpopulation, $I_{NaP}$ with its favorable voltage-dependent and kinetic autorhythmic properties– is not proposed to be a basic biophysical mechanism for rhythm generation in the burstlet theory (*Del Negro et al., 2018*).

We emphasize that the above discussions regarding the role of $I_{NaP}$ pertain to the excitatory circuits in the isolated pre-BötC including in more mature rodent experimental preparations in situ where inspiratory rhythm generation has also been shown to be dependent on $I_{NaP}$ (*Smith et al., 2007*). The analysis is more complex when the pre-BötC is embedded within interacting respiratory circuits in the intact nervous system generating the full complement of inspiratory and expiratory phase activity (*Lindsey et al., 2012*; *Ramirez and Baertsch, 2018a*; *Richter and Smith, 2014*), where rhythmogenesis is tightly controlled by inhibitory circuit interactions, including via local inhibitory circuits in the pre-BötC (*Harris et al., 2017*; *Ausborn et al., 2018*; *Baertsch et al., 2018*; *Ramirez and Baertsch, 2018a*), and the contribution of $I_{NaP}$ kinetic properties alone in setting the timing of inspiratory oscillations is diminished (*Smith et al., 2007*; *Richter and Smith, 2014*; *Rubin et al., 2009b*). Extending our analysis to consider inhibitory circuit interactions with the excitatory subpopulations generating oscillation frequency and amplitude that we propose will provide additional insight into biophysical mechanisms controlling these two processes.

## Conclusions

Based on our computational model, distinct biophysical mechanisms are involved in generating the rhythm and amplitude of inspiratory oscillations in the isolated pre-BötC excitatory circuits. According to this model, inspiratory rhythm generation arises from a group of $I_{NaP}$-dependent excitatory neurons, including cells with intrinsic oscillatory bursting properties, that form a rhythmogenic kernel. Rhythmic synaptic drive from these neurons triggers post-synaptic calcium transients,

$I_{CAN}$ activation, and subsequent membrane depolarization which drives rhythmic bursting in the rest of the population of inspiratory neurons. We showed that activation of $I_{CAN}$ by synaptically-driven calcium influx functions as a mechanism that amplifies the excitatory synaptic input to generate the inspiratory drive potential and population activity amplitude in these non-rhythmogenic neurons. Consequently, reduction of $I_{CAN}$ causes a robust decrease in overall network activity amplitude via de-recruitment of these burst amplitude-generating neurons without substantial perturbations of the inspiratory rhythm. Thus, $I_{CAN}$ plays a critical role in generating the amplitude of rhythmic population activity, which is consistent with the results from experimental inhibition of $I_{CAN}$/TRPM4 channels (*Peña et al., 2004*; *Koizumi et al., 2018*). Our model provides a theoretical explanation for these experimental results and new insights into the biophysical operation of pre-BötC excitatory circuits. The theoretical framework that we have developed here should provide the bases for further exploration of biophysical mechanisms operating in the mammalian respiratory oscillator.

## Materials and methods

### Model description

The model describes a network of $N = 100$ synaptically coupled excitatory neurons. Simulated neurons are comprised of a single compartment described using a Hodgkin Huxley formalism. For each neuron, the membrane potential $V_m$ is given by the following current balance equation:

$$C_m \frac{dV_m}{dt} + I_{Na} + I_K + I_{Leak} + I_{NaP} + I_{CAN} + I_{Ca} + I_{Syn} = 0$$

where $C_m$ is the membrane capacitance, $I_{Na}$, $I_K$, $I_{Leak}$, $I_{NaP}$, $I_{CAN}$, $I_{Ca}$ and $I_{Syn}$ are ionic currents through sodium, potassium, leak, persistent sodium, calcium activated non-selective cation, voltage-gated calcium, and synaptic channels, respectively. Description of these currents, synaptic interactions, and parameter values are taken from *Jasinski et al. (2013)*. The channel currents are defined as follows:

$$I_{Na} = \bar{g}_{Na} \cdot m_{Na}^3 \cdot h_{Na} \cdot (V_m - E_{Na})$$

$$I_K = \bar{g}_K \cdot m_K^4 \cdot (V_m - E_K)$$

$$I_{Leak} = \bar{g}_{Leak} \cdot (V_m - E_{Leak})$$

$$I_{NaP} = \bar{g}_{NaP} \cdot m_{NaP} \cdot h_{NaP} \cdot (V_m - E_{Na})$$

$$I_{CAN} = \bar{g}_{CAN} \cdot m_{CAN} \cdot (V_m - E_{CAN})$$

$$I_{Ca} = \bar{g}_{Ca} \cdot m_{Ca} \cdot h_{Ca} \cdot (V_m - E_{Ca})$$

$$I_{Syn} = g_{Syn} \cdot (V_m - E_{Syn})$$

where $\bar{g}_i$ is the maximum conductance, $E_i$ is the reversal potential, $m_i$ and $h_i$ are voltage dependent gating variables for channel activation and inactivation, respectively, and $i \in \{Na, K, Leak, NaP, CAN, Ca, Syn\}$. The parameters $\bar{g}_i$ and $E_i$ are given in *Table 1*.

For $I_{Na}$, $I_K$, $I_{NaP}$, and $I_{Ca}$, the dynamics of voltage-dependent gating variables $m_i$, and $h_i$ are defined by the following differential equation:

$$\tau_\eta(V) \cdot \frac{d\eta}{dt} = \eta_\infty(V) - \eta; \ \eta \in \{m_i, h_i\}$$

where steady state activation/inactivation $\eta_\infty$ and time constant $\tau_\eta$ are given by:

$$\eta_\infty(V) = \left(1 + e^{-\left(V - V_{\eta_{1/2}}\right)/k_\eta}\right)^{-1}$$

$$\tau_\eta(V) = \tau_{\eta_{max}} / \cosh\left(\left(V - V_{\tau\eta_{1/2}}\right) / k_{\tau_\eta}\right).$$

For the voltage-gated potassium channel, steady state activation $m_{K\infty}(V)$ and time constant $\tau_{mK}(V)$ are given by:

$$m_{K\infty}(V) = \frac{\alpha_\infty(V)}{\alpha_\infty(V) + \beta_\infty(V)}$$

$$\tau_{mK}(V) = 1/(\alpha_\infty(V) + \beta_\infty(V))$$

where

$$\alpha_\infty(V) = A_\alpha \cdot (V + B_\alpha)/(1 - exp(-(V + B_\alpha)/\kappa_\alpha))$$

$$\beta_\infty(V) = A_\beta \cdot exp\left(-\left(V + B_\beta\right)/\kappa_\beta\right).$$

The parameters $V_{\eta_{1/2}}$, $V_{\tau\eta_{1/2}}$, $\kappa_\eta$, $\kappa_{\tau\eta}$ $\tau_{\eta max}$, $A_\alpha$, $A_\beta$, $B_\alpha$, $B_\beta$, $\kappa_\alpha$, and $\kappa_\beta$ are given in **Table 1**. $I_{CAN}$ activation is dependent on the intracellular calcium concentration $[Ca]_{in}$ and is given by:

$$m_{CAN} = 1/\left(1 + \left(Ca_{1/2}/[Ca]_{in}\right)^n\right).$$

The parameters $Ca_{1/2}$ and $n$, given in **Table 1**, represent the half-activation calcium concentration and the Hill Coefficient, respectively.

Calcium enters the neurons through voltage-gated calcium channels ($Ca_V$) and/or synaptic channels ($Ca_{Syn}$), where a percentage ($P_{Ca}$) of the synaptic current ($I_{Syn}$) is assumed to consist of $Ca^{2+}$ ions. A calcium pump removes excess calcium with a time constant $\tau_{Ca}$ and sets the minimum calcium concentration $Ca_{min}$. The dynamics of $[Ca]_{in}$ is given by the following differential equation:

$$\frac{d[Ca]_{in}}{dt} = -\alpha_{Ca}\left(I_{Ca} + P_{Ca} \cdot I_{syn}\right) - \left([Ca]_{in} - Ca_{min}\right)/\tau_{Ca}.$$

The parameters $\alpha_{Ca}$ is a conversion factor relating current and rate of change in $[Ca]_{in}$, see **Table 1** for parameter values.

The synaptic conductance of the $i^{th}$ neuron ($g_{Syn}^i$) in the population is described by the following equation:

$$g_{Syn}^i = g_{Tonic} + \sum_{j,n} w_{ji} \cdot C_{ji} \cdot H\left(t - t_{j,n}\right) \cdot e^{-\left(t - t_{j,n}\right)/\tau_{syn}}$$

where $w_{ji}$ is the weight of the synaptic connection from cell $j$ to cell $i$, $C$ is a connectivity matrix ($C_{ji} = 1$ if neuron $j$ makes a synapse on neuron $i$, and $C_{ji} = 0$ otherwise), $H(.)$ is the Heaviside step function, $t$ is time, $\tau_{Syn}$ is the exponential decay constant and $t_{j,n}$ is the time at which an action potential $n$ is generated in neuron $j$ and reaches neuron $i$.

To account for heterogeneity of neuron properties within the network, the persistent sodium current conductance, $\bar{g}_{NaP}$, for each neuron was assigned randomly based on a uniform distribution over the range $[0.0, 5.0]$ $nS$ which is consistent with experimental measurements (**Rybak et al., 2003**; **Koizumi and Smith, 2008**; **Koizumi and Smith, 2008**). We also uniformly distributed $\bar{g}_{CAN}$ over the range $[0.5, 1.5]$ $nS$, however, simulation results did not depend on whether we used such a distribution, or assigned $\bar{g}_{CAN}$ for all neurons to the same value of $1.0$ $nS$, which is the mean of this distribution. In simulations where $\bar{g}_{CAN}$ was varied, we multiplied $\bar{g}_{CAN}$ for each neuron by the same factor. This factor was used as a control parameter for all such simulations, and shown as a percentage of the baseline $\bar{g}_{CAN}$ or as the mean $\bar{g}_{CAN}$ values for the population in figures. The weight of each synaptic connection was uniformly distributed over the range $w_{ji} \in [0, W_{max}]$ where $W_{max}$ ranged from $0.0$ to $1.0$ $nS$ depending on the network connectivity and specific simulation. The elements of the network connectivity matrix, $C_{ji}$, are randomly assigned values of 0 or 1 such that the probability of any connection between neuron $j$ and neuron $i$ being 1 is equal to the network connection probability $P_{Syn}$.

**Table 1.** Model parameter values.

The channel kinetics, intracellular Ca$^{2+}$ dynamics and the corresponding parameter values, were derived from previous models (see *Jasinski et al., 2013*) and the references therein.

| Channel | Parameters |
|---|---|
| $I_{Na}$ | $\bar{g}_{Na} = 150.0\ nS,\ E_{Na} = 55.0\ mV,$ <br> $V_{m_{1/2}} = -43.8\ mV,\ k_m = 6.0\ mV,$ <br> $V_{\tau m_{1/2}} = -43.8\ mV,\ k_{\tau_m} = 14.0\ mV,\ \tau_{m_{max}} = 0.25\ ms,$ <br> $V_{h_{1/2}} = -67.5\ mV,\ k_h = -10.8\ mV,$ <br> $V_{\tau h_{1/2}} = -67.5\ mV,\ k_{\tau_h} = 12.8\ mV,\ \tau_{h_{max}} = 8.46\ ms$ |
| $I_K$ | $\bar{g}_K = 160.0\ nS,\ E_K = -94.0\ mV,$ <br> $A_\alpha = 0.01,\ B_\alpha = 44.0\ mV,\ \kappa_\alpha = 5.0\ mV$ <br> $A_\beta = 0.17,\ B_\beta = 49.0\ mV,\ \kappa_\beta = 40.0\ mV$ |
| $I_{Leak}$ | $\bar{g}_{Leak} = 2.5\ nS,\ E_{Leak} = -68.0\ mV$ |
| $I_{NaP}$ | $\bar{g}_{NaP} \in [0.0, 5.0]\ nS,$ <br> $V_{m_{1/2}} = -47.1\ mV,\ k_m = 3.1\ mV,$ <br> $V_{\tau m_{1/2}} = -47.1\ mV,\ k_{\tau_m} = 6.2\ mV,\ \tau_{m_{max}} = 1.0\ ms,$ <br> $V_{h_{1/2}} = -60.0\ mV,\ k_h = -9.0\ mV,$ <br> $V_{\tau h_{1/2}} = -60.0\ mV,\ k_{\tau_h} = 9.0\ mV,\ \tau_{h_{max}} = 5000\ ms$ |
| $I_{CAN}$ | $\bar{g}_{CAN} \in [0.5, 1.5]\ nS,\ E_{CAN} = 0.0\ mV,$ <br> $Ca_{1/2} = 0.00074\ mM,\ n = 0.97$ |
| $I_{Ca}$ | $\bar{g}_{Ca} = 0.01\ nS,\ E_{Ca} = R \cdot T/F \cdot ln([Ca]_{out}/[Ca]_{in}),$ <br> $R = 8.314\ J/(mol \cdot K),\ T = 308.0\ K,$ <br> $F = 96.485\ kC/mol,\ [Ca]_{out} = 4.0\ mM$ <br> $V_{m_{1/2}} = -27.5\ mV,\ k_m = 5.7\ mV,\ \tau_m = 0.5\ ms,$ <br> $V_{h_{1/2}} = -52.4\ mV,\ k_h = -5.2\ mV,\ \tau_h = 18.0\ ms$ |
| $Ca_{in}$ | $\alpha_{Ca} = 2.5 \cdot 10^{-5}\ mM/fC,\ P_{Ca} = 0.01,\ Ca_{min} = 1.0 \cdot 10^{-10}mM,\ \tau_{Ca} = 50.0\ ms$ |
| $I_{Syn}$ | $g_{Tonic} = 0.31\ nS,\ E_{Syn} = -10.0\ mV,\ \tau_{Syn} = 5.0\ ms$ |

DOI: https://doi.org/10.7554/eLife.41555.013

We varied the connection probability over the range $P_{Syn} \in [0.05, 1.0]$, however, a value of $P_{Syn} = 0.05$ was used in most simulations.

## Data analysis and definitions

The time of an action potential was defined as when the membrane potential of a neuron crosses $-35mV$ in a positive direction. The network activity amplitude and frequency were determined by identifying peaks and calculating the inverse of the interpeak interval in histograms of network spiking. Network histograms of the population activity were calculated as the number of action potentials generated by all neurons per 50 $ms$ bin per neuron with units of $spikes/s$. The number of recruited neurons is defined as the peak number of neurons that spiked at least once per bin during a network burst. The average spike frequency of recruited neurons is defined as the number of action potentials per bin per recruited neuron with units of $spikes/s$. The average network resting membrane potential was defined as the average minimum value of $V_m$ in a 500 $ms$ window following a network burst. The average inactivation of the persistent sodium current at the start of each burst was defined by the maximum of the average value of $h_{NaP}$ in a 500-ms window before the peak of each network burst. The average inactivation of the persistent sodium current at the end of each burst was defined by the maximum of the average value of $h_{NaP}$ in a 500-ms window after the peak of each network burst. Synaptic strength is defined as the number of neurons in the network multiplied by the connection probability multiplied by the average weight of synaptic connections $(N \cdot P_{Syn} \cdot \frac{1}{2} W_{max})$. Pacemaker neurons were defined as neurons that continue bursting intrinsically after complete synaptic blockade. Follower neurons were defined as neurons that become silent after complete synaptic blockade. The inspiratory drive potential is defined as the envelope of depolarization that occurs in neurons during the inspiratory phase of the network oscillations (*Morgado-Valle et al., 2008*).

## Characterization of $I_{CAN}$ in regulating network activity amplitude and frequency in $Ca_V$ and $Ca_{Syn}$ models

To characterize the role of $I_{CAN}$ in regulation of network activity amplitude and frequency we slowly increased the conductance ($\bar{g}_{CAN}$) in our simulations from zero until the network transitioned from a rhythmic bursting to a tonic (non-bursting) firing regime. To ensure that the effect(s) are robust, these simulations were repeated over a wide range of synaptic weights, synaptic connection probabilities, and strengths of the intracellular calcium transients from $Ca_V$ or $Ca_{Syn}$ sources. Changes in network activity amplitude were further examined by plotting the number of recruited neurons and the average action potential frequency of recruited neurons versus $\bar{g}_{CAN}$.

## Simulated pharmacological manipulations

In simulations that are compared with experimental data, both $Ca_V$ and $Ca_{Syn}$ calcium sources are included. Pharmacological blockade of $I_{CAN}$ was simulated by varying the conductance, $\bar{g}_{CAN}$ according to a decaying exponential function

$$\bar{g}_{CAN}(t) = g_{CAN}^{max} - \gamma_{block} \cdot \left(1 - e^{-t/\tau_{block}}\right).$$

The percent block $\gamma_{block}$, decay constant $\tau_{block}$ and the maximum $I_{CAN}$ conductance $g_{CAN}^{max}$ were adjusted to match the experimental changes in network amplitude. The synaptic weight of the network was chosen such that at $\bar{g}_{CAN} = 0$ the network activity amplitude was close to 20% of maximum. To reduce the computational time, the duration of $I_{CAN}$ block simulations was one tenth of the total of experimental durations. For comparison, the plots of normalized change in amplitude and frequency of the simulations were stretched over the same time-period as experimental data. Increasing the simulation time had no effect on our results (data not shown).

## Comparison with calcium imaging data

To allow comparisons with network and cellular calcium imaging data, we analyzed rhythmic calcium transients from our simulations. Single cell calcium signals are represented by $[Ca]_i$. The network calcium signal was calculated as the average intracellular calcium concentration in the network ($\sum\limits_{1}^{N}[Ca]_i/N$).

## Integration methods

All simulations were performed locally on an 8-core Linux-based operating system or on the high-performance computing cluster Biowulf at the National Institutes of Health. Simulation software was custom written in C++. Numerical integration was performed using the exponential Euler method with a fixed step-size ($\Delta t$) of $0.025\,ms$. In all simulations, the first 50 s of simulation time was discarded to allow for the decay of any initial condition-dependent transients.

## Acknowledgements

This work was supported in part by the Jayne Koskinas and Ted Giovanis Foundation for Health and Policy, the Intramural Research Program of the National Institutes of Health (NIH), National Institute of Neurological Disorders and Stroke (NINDS), and NIH Grants R01 AT008632 and U01 EB021960.

## Additional information

### Funding

| Funder | Grant reference number | Author |
|---|---|---|
| The Jayne Koskinas Ted Giovanis Foundation for Health and Policy | | Ryan S Phillips |
| National Institute of Neurological Disorders and Stroke | Intramural Research Program of the National Institutes of Health | Ryan S Phillips Tibin T John Hidehiko Koizumi Jeffrey C Smith |

| National Institutes of Health | R01 AT008632 | Yaroslav I Molkov |
| National Institutes of Health | U01 EB021960 | Yaroslav I Molkov |

The funders had no role in study design, data collection and interpretation, or the decision to submit the work for publication.

## Author contributions
Ryan S Phillips, Yaroslav I Molkov, Jeffrey C Smith, Conceptualization, Data curation, Formal analysis, Writing—original draft, Writing—review and editing; Tibin T John, Hidehiko Koizumi, Data curation, Formal analysis

## Author ORCIDs
Ryan S Phillips ⬛ http://orcid.org/0000-0002-8570-2348
Tibin T John ⬛ http://orcid.org/0000-0002-6825-7166
Yaroslav I Molkov ⬛ http://orcid.org/0000-0002-0862-1974
Jeffrey C Smith ⬛ http://orcid.org/0000-0002-7676-4643

## Decision letter and Author response
Decision letter https://doi.org/10.7554/eLife.41555.017
Author response https://doi.org/10.7554/eLife.41555.018

## Additional files

### Supplementary files
• Source code 1. Phillips et al. model source code.
DOI: https://doi.org/10.7554/eLife.41555.014
• Transparent reporting form
DOI: https://doi.org/10.7554/eLife.41555.015

### Data availability
All data in this study are generated by computational simulations. All model parameters and equations are included in the manuscript and source code is included with this submission.

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
