## [Decision Letter]

Thank you for submitting your article "Biophysical mechanisms in the mammalian respiratory oscillator re-examined with a new data-driven computational model" for consideration by *eLife*. Your article has been reviewed by two peer reviewers, including Jan-Marino Ramirez as the Reviewing Editor and Reviewer #1, and the evaluation has been overseen by Ronald Calabrese as the Senior Editor. The following individual involved in review of your submission has agreed to reveal their identity: Guillaume Drion (Reviewer #3).

The reviewers have discussed the reviews with one another and the Reviewing Editor has drafted this decision to help you prepare a revised submission.

Summary:

The manuscript by Phillips et al. explores the interplay between two inward currents (*I_CAN_* and *I_NaP_*) that have been implicated in respiratory rhythm generation within the pre-Bötzinger complex. Using computational modeling the authors carefully characterize the relative rhythmogenic contributions of each of these currents, which are dynamic, can differ in individual neurons and can change when one current is blocked, or when the intracellular calcium concentration changes. The authors use a computational model made of 100 synaptically coupled excitatory neurons with random connectivity and random *I_NaP_* conductance (and *I_CAN_* conductance?). Each neuron model is conductance-based and includes 5 voltage-gated conductances, 1 leak conductance, intracellular calcium dynamics and synaptic currents. The paper compares two different model configurations: a configuration where voltage-gated calcium channels are the main source of calcium for *I_CAN_* activation (*Ca_V_* network) and a configuration where *I_CAN_* is activated by calcium coming from synaptically activated sources (*Ca_Syn_* network). First, analyses on the role of *I_CAN_* conductance on network amplitude and frequency in both configurations and comparison with recent experimental data suggest that *I_CAN_* activation mostly relies on synaptically-activated calcium sources, which confirms previous studies. Second, the paper shows that *I_NaP_* is critical for network rhythmic activity: it imbues some neurons of the network with spontaneous bursting capabilities, forming an *I_NaP_*-dependent rhythmogenic kernel. We find this study important, in part because the dynamic nature could also explain why there are so many discrepancies in the field regarding the role of these two inward currents.

Title:

In the title, the paper mentions the use of a "data-driven computational model". I was therefore expecting to see a model whose parameters were set according to experimental data, which is not the case in this paper (or at least not a contribution of this paper). I understand that the model structure was designed to investigate the mechanisms that could explain recent experimental data, which I appreciate, but calling the model "data-driven" seems misleading to me (even if data are used to quantify the contributions of *Ca_V_* and *Ca_Syn_* in Figure 7).

Essential revisions:

1) The authors focused only on the fact that blocking *I_CAN_* has a small frequency effect but large amplitude effect, which is something that has been previously reported in 2004 (Pena et al., 2004), and which these researchers reproduced in 2018. The authors don't address the finding that blocking *I_NaP_* alone has also no effect on the frequency (Pena et al., 2004), while others show that blocking *I_NaP_* alone can block rhythmogenesis under control conditions. By contrast Pena et al. found that this is only the case in severe hypoxia. The modeling could explain these different experimental findings. For example the intracellular calcium concentrations or the network state could differ between different labs. Unfortunately, the authors don't pitch the paper like this, which seems like a missed opportunity. Thus, the authors should try to be less polarizing and report data as is (and as detailed below in more detail). There is no need to try to push the agenda that only *I_NaP_* is rhythmogenic when we know that *I_CAN_* has a big role in rhythmogenesis by regulating synaptic excitatory transmission.

2) The authors should add a discussion regarding the neuron model and the robustness of the network activity. First, the model generates bursts of action potentials having an amplitude of around 30-40mV, as shown in Figure 4 and Figure 10. But in the experimental data of Koizumi et al., the amplitude of action potentials is more about roughly 80-100 mV (Figure 3 of the paper). Why is there such a big difference between the two? The spike height is indeed often used as a major criterion to assess the validity of a computational model at the cellular level. We agree that it is not a critical feature of the model used in this paper, but the issue deserves to be discussed. In particular, what happens to the bursting pattern if you attempt to reach a physiologically plausible spike height by increasing the transient sodium conductance (which might also require an increase in delayed-rectifier potassium conductance)? We suspect that bursting would be very fragile to an increase in Na conductance, due to the specific dynamical mechanisms at play in the model.

3) Along the same lines, Figure 4A suggests that the region of network bursting is quite narrow as compared to the region of network tonic spiking (especially if one increases the parameter range). What is the robustness of this bursting region against variability/heterogeneity in the other conductance values? Do you think that this result implies that physiological mechanisms need to tightly regulate the values of the conductances to maintain a viable network bursting activity (for instance a specific balance between synaptic strength and *I_CAN_* conductance), or could it be possible that this fragility is an artifact of the computational model?

4) The authors should succinctly describe the model at the beginning of the Results section. Indeed, it was difficult for us to understand the configuration of the model without having to go through the Materials and methods section, which is at the end of the manuscript and very detailed. It is important that the reader has a clear view of how neurons are modeled (ion channels), how many neurons constitute the circuit, how you compute "circuit activity amplitude", etc. before going through the analyses. Please also make sure to define every specific term when it is introduced. For instance, when describing how network probability is computed, *W_max_* is not defined, and it cannot be easily related to the "average weight of synaptic connections" without again going through the Materials and methods section. Also, you mention in the Abstract that *I_CAN_* conductance is distributed randomly, but not in the Materials and methods section. Which one is correct?

---

## [Author Response]

Title:In the title, the paper mentions the use of a "data-driven computational model". I was therefore expecting to see a model whose parameters were set according to experimental data, which is not the case in this paper (or at least not a contribution of this paper). I understand that the model structure was designed to investigate the mechanisms that could explain recent experimental data, which I appreciate, but calling the model "data-driven" seems misleading to me (even if data are used to quantify the contributions of Ca_V_ and Ca_Syn_ in Figure 7).

We do not see the problem for the reasons that the reviewers state and do not think that calling the model “data-driven” is misleading. The modeling was entirely motivated by new data requiring a theoretical and conceptual explanation including model fits to some of the key data.

Essential revisions:1) The authors focused only on the fact that blocking I_CAN_ has a small frequency effect but large amplitude effect, which is something that has been previously reported in 2004 (Pena et al., 2004), and which these researchers reproduced in 2018. The authors don't address the finding that blocking I_NaP_ alone has also no effect on the frequency (Pena et al., 2004), while others show that blocking I_NaP_ alone can block rhythmogenesis under control conditions. By contrast Pena et al. found that this is only the case in severe hypoxia. The modeling could explain these different experimental findings. For example the intracellular calcium concentrations or the network state could differ between different labs. Unfortunately, the authors don't pitch the paper like this, which seems like a missed opportunity. Thus, the authors should try to be less polarizing and report data as is (and as detailed below in more detail). There is no need to try to push the agenda that only I_NaP_ is rhythmogenic when we know that I_CAN_ has a big role in rhythmogenesis by regulating synaptic excitatory transmission.

We focused on independence of the amplitude and the frequency of oscillations which is mathematically counterintuitive and represents a major qualitative effect observed experimentally that must be indicating something important about the dynamical/biophysical organization and operation of the isolated pre-BötC network. This problem has not been previously addressed. Reference to Pena et al. (2004) was added in the Introduction and throughout. While amplitude modulation associated with experimental attenuation of *I_CAN_* conductance has been reported, these and other studies referenced have emphasized the potential role of *I_CAN_* in intrinsic neuronal bursting and rhythm generation by pre-BötC circuits, justifiably since calcium-dependent/*I_CAN_* intrinsic neuronal bursting has been observed, starting with the studies of Thoby-Brisson and Ramirez (2001) and Pena et al. (2004). Yes, our model by careful analysis of the sources of calcium flux activating *I_CAN_*, can explain the observations of Pena et al. of large perturbations of population activity amplitude with relatively smaller perturbations of oscillation frequency of the isolated pre-BötC in vitro, as well as similar results from the extensive experimental analysis of Koizumi et al. (2018) directly addressing this problem.

“The authors don't address the finding that blocking I_NaP_ alone has also no effect on the frequency (Pena et al., 2004), while others show that blocking I_NaP_ alone can block rhythmogenesis under control conditions. By contrast Pena et al. found that this is only the case in severe hypoxia. The modeling could explain these different experimental findings. For example the intracellular calcium concentrations or the network state could differ between different labs. Unfortunately, the authors don't pitch the paper like this, which seems like a missed opportunity. Thus, the authors should try to be less polarizing and report data as is (and as detailed below in more detail).”

Our statements are based largely on our published experimental results, including that application of riluzole in concentrations used by Pena et al. always led to cessation of the rhythm in rat slices under control conditions, which has also been reported in mouse slice as referenced (Toporikova et al., 2015). We have now noted in the Introduction “although some studies suggest that block of both *I_NaP_* and *I_CAN_* are necessary to disrupt rhythmogenesis in vitro (Pena et al., 2004).” We have not attempted to solve this particular problem here, because of the focus in the modeling on tackling the problem of calcium dynamics and *I_CAN_*activation, but we have previously discussed potential differences in experimental preparations including network state that could possibly explain such differences in experimental results from a theoretical perspective in Jasinski et al. (2013) which we have now discussed (see Discussion section “Role of I_CAN_ in the pre-BötC Respiratory Network”). We have also discussed how our model can explain the small perturbations of oscillation frequency accompanying large reductions in inspiratory population activity that have been observed experimentally with *I_CAN_* block by some investigators (e.g., Pena et al., 2004). We think that we have achieved a balanced and accurate presentation of experimental data throughout the present paper as the reviewers advise, and we have made revisions in response to the reviewers’ specific suggestions in this regard. The goal of the present study was to provide an explanation and formulate implications for the fact that the excitatory pre-BötC population is capable of varying the amplitude of its oscillations essentially independent of the frequency, which is analytically challenging, and our model formulation can potentially serve as a basis for resolving various experimental discrepancies with additional focused analyses.

“There is no need to try to push the agenda that only I_NaP_ is rhythmogenic when we know that I_CAN_ has a big role in rhythmogenesis by regulating synaptic excitatory transmission.”

It is indeed the main conclusion of our study that *I_CAN_* contributes critically to excitatory synaptic interactions rather than intrinsic rhythmogenic properties of pre-BötC excitatory circuits. This is what explains why the network activity amplitude is very sensitive to manipulations of *I_CAN_* conductance while the oscillation frequency is not. We note that complete blockade of *I_CAN_* in our model can abolish synchronized network oscillations due to weakened excitatory synaptic transmission, which results in neuronal de-recruitment and desynchronization of the network, particularly when synaptic strength is low. So *I_CAN_*plays an important role in network activity synchronization but this is not rhythmogenesis per se. We have now noted this role of *I_CAN_* in synchronization in the Discussion section “Role of I_CAN_ in the pre-BötC Respiratory Network”.

2) The authors should add a discussion regarding the neuron model and the robustness of the network activity. First, the model generates bursts of action potentials having an amplitude of around 30-40mV, as shown in Figure 4 and Figure 10. But in the experimental data of Koizumi et al., the amplitude of action potentials is more about roughly 80-100 mV (Figure 3 of the paper). Why is there such a big difference between the two? The spike height is indeed often used as a major criterion to assess the validity of a computational model at the cellular level. We agree that it is not a critical feature of the model used in this paper, but the issue deserves to be discussed. In particular, what happens to the bursting pattern if you attempt to reach a physiologically plausible spike height by increasing the transient sodium conductance (which might also require an increase in delayed-rectifier potassium conductance)? We suspect that bursting would be very fragile to an increase in Na conductance, due to the specific dynamical mechanisms at play in the model.

While it is certainly true that neuronal bursting capability could be affected by an increase in the transient sodium conductance without adjustments to other conductances, the bursting neuron model that we employ represents a stable bursting solution with the specific dynamical mechanisms at play. Indeed, complete biophysically accurate modeling of the respiratory neurons was not the goal of this study. Network synaptic interactions formulated in our model depend on times of spike occurrence and not on spike shape, height, or duration. Network activity is also characterized in terms of the number of generated spikes and not individual voltage dynamics. This allows us to claim that the results presented hold regardless of spiking characteristics. We, therefore, do not think that the discussion of the role of intrinsic spike properties would be especially important, which the reviewers seem to acknowledge. We also note that the transient sodium and delayed-rectifier potassium channel conductance (gKdr) dynamics in the current study are borrowed from previously published pre-BötC neuron models (Rybak et al., 2007; Smith et al., 2007; Jasinski et al., 2013). As in these previous studies, the spike height in the current model is approximately 45 mV which is smaller than the spikes shown in (Koizumi et al., 2018), although the spike height in the current model is within the range (35-80mV) of those shown in the numerous publications from other labs (e.g., Thoby-Brisson et al., 2001; Pena et al., 2004; Del Negro et al., 2005; Morgado-Valle et al., 2008; Ben-Mabrouk, 2010). We note that in the original model formulation of Butera et al. (1999a), *I_NaP_*-dependent rhythmic bursting occurs with larger amplitude spikes than the present model, so bursting solutions exist with spike heights closer to those observed by Koizumi et al. We also note that in Figure 4E, F the spikes shown were deliberately truncated, as originally noted in the figure legend. Overall, we have attempted to demonstrate the robustness of the model network activity, particularly with respect to effects of changes in g_CAN_, in Figures 5 and 6. We have now indicated in the text the dependence of the rhythmogenic kernel on other factors such as leak conductance and gKdr in the section of the Discussion on “*I_NaP_*-dependent rhythmogenic kernel” to give the readers some sense for what determines intrinsic rhythmic bursting in the model.

3) Along the same lines, Figure 4A suggests that the region of network bursting is quite narrow as compared to the region of network tonic spiking (especially if one increases the parameter range). What is the robustness of this bursting region against variability/heterogeneity in the other conductance values? Do you think that this result implies that physiological mechanisms need to tightly regulate the values of the conductances to maintain a viable network bursting activity (for instance a specific balance between synaptic strength and I_CAN_ conductance), or could it be possible that this fragility is an artifact of the computational model?

We thank the reviewers for pointing out this issue. We added new text to the section (“Manipulating g-CAN in the CaSyn model is qualitatively equivalent to changing the strength of synaptic interactions”) describing Figure 4A to clarify the following. Because *I_CAN_* acts as a synaptic amplifier, the effective strength of synaptic interactions is roughly proportional to a product of the synaptic weight (SynW, shown as Synaptic Strength in Figure 4) andg-CAN. A transition from bursting to tonic spiking occurs when this effective excitation exceeds a certain critical value Wc in any imaginable network with recurrent excitation. That is why the bifurcation curve representing this boundary (between yellow and black) on the parameter plane (g-CAN, SynW) looks like a hyperbola g-CAN x SynW=Wc. Indeed, the stronger the basic synaptic interactions, the less room *I_CAN_* has to further facilitate them before the transition to the tonic regime occurs. So, all that Figure 4A implies is that the product of SynW and g-CAN should not exceed a certain value for the bursting to persist. It does not imply any specific balance between the two or non-robustness of the bursting regime.

4) The authors should succinctly describe the model at the beginning of the Results section. Indeed, it was difficult for us to understand the configuration of the model without having to go through the Materials and methods section, which is at the end of the manuscript and very detailed. It is important that the reader has a clear view of how neurons are modeled (ion channels), how many neurons constitute the circuit, how you compute "circuit activity amplitude", etc. before going through the analyses. Please also make sure to define every specific term when it is introduced. For instance, when describing how network probability is computed, W_max_ is not defined, and it cannot be easily related to the "average weight of synaptic connections" without again going through the Materials and methods section. Also, you mention in the Abstract that I_CAN_ conductance is distributed randomly, but not in the Materials and methods section. Which one is correct?

A succinct model description has now been added to the beginning of the Results sections and definitions for circuit activity amplitude and other terms such as synaptic strength have been added to the text when they are introduced. In the model, we also distributed *I_CAN_* conductance (g-CAN) based on a uniform distribution and this is now indicated in Table 1 and described in the Materials and methods. We actually found that the simulation results were not dependent on such a distribution as now stated in Materials and methods, and for the representative simulations presented in the figures, g-CAN values indicated in the figures represent the mean values for the simulated populations. Information about random ionic and synaptic conductance distributions has been removed from the Abstract, where such information is not really needed.